



# The Ozone Monitoring Instrument: Overview of twelve years in space

Pieternel Levelt[1,2], Joanna Joiner[3], Johanna Tamminen[4], Pepijn Veefkind[1,2], Pawan K. Bhartia[3], Deborah Stein Zweers[1], Bryan N. Duncan[3], David G. Streets[5], Henk Eskes[1], Ronald van der A[1], Chris McLinden[6], Vitali Fioletov[6], Simon Carn[7], Jos de Laat[1], Matthew DeLand[8], Sergey Marchenko[8], Richard McPeters[3], Jerald Ziemke[3,9], Dejian Fu[10], Xiong Liu[11], Kenneth Pickering[3,12], Arnoud Apituley[1], Gonzalo Gonzáles Abad[11], Antti Arola[4], Folkert Boersma[1,13], Christoph Chan Miller[11], Kelly Chance[11], Martin de Graaf[1], Janne Hakkarainen[4], Seppo Hassinen[4], Iolanda Ialongo[4], Quintus Kleipool[1], Nickolay Krotkov[3], Can Li[12], Lok Lamsal[14], Paul Newman[3], Caroline Nowlan[11], Raid Suileiman[11], Lieuwe Gijsbert Tilstra[1], Omar Torres[3], Huiqun Wang[11], Krzysztof Wargan[3,8]

[1]Royal Netherlands Meteorological Institute, De Bilt, 3731 GA, The Netherlands
[2]Faculty of Civil Engineering and Geosciences, Delft, 2628 CN, The Netherlands
[3]NASA Goddard Space Flight Center, Greenbelt, Maryland, 20771, USA
[4]Finnish Meteorological Institute, Helsinki, FI-00101, Finland
[5]Energy Energy Systems Division, Argonne National Laboratory, Argonne, Illinois, 60439, USA
[6]Air Quality Research Division, Environment and Climate Change Canada, Toronto, M3H 5T4, Canada
[7]Geological and Mining Engineering and Sciences, Michigan Technological University, Houghton, Michigan, 49931, USA
[8]Science Systems and Applications, Inc., Lanham, Maryland, 20706, USA
[9]Morgan State University, Baltimore, Maryland, 21251, USA
[10]NASA Jet Propulsion Laboratory, Pasadena, California, 91109, USA
[11]Harvard-Smithsonian Center for Astrophysics, Cambridge, Massachusetts, 02138, USA
[12]Department of Atmospheric and Oceanic Sciences, University of Maryland, College Park, Maryland, 20742, USA
[13]Department of Environmental Sciences, Wageningen University, Wageningen, 6708 PB, The Netherlands
[14]Universities Space Research Association, Columbia, MD, 21046 USA

*Correspondence to:* Pieternel Levelt (levelt@knmi.nl)

**Abstract.** This overview paper highlights the successes of the Ozone Monitoring Instrument (OMI) spanning more than 12 years of the OMI data record. Data from OMI has been used in a wide range of applications. Due to its unprecedented spatial resolution, in combination with daily global coverage, OMI plays a unique role in measuring trace gases important for the ozone layer, air quality and climate change, including new research findings using these satellite data. Due to the operational Very Fast Delivery (VFD) (direct readout) and Near Real Time (NRT) availability of the data, OMI also plays an important role in the early development of operational services in the atmospheric chemistry domain.

## 1 Introduction

On July 15, 2004, the Dutch-Finnish Ozone Monitoring Instrument (OMI) was launched on board the US National Aeronautics and Space Administration (NASA) Earth Observing System (EOS) Aura spacecraft. After more than 12 years of operations, OMI still continues to provide unique data for atmospheric research and applications. In this overview paper that is part of the ACP/AMT OMI Special Issue, we aim to highlight OMI's exceptional instrument design features, as well as some of OMI's accomplishments. Detailed results can be found in other contributions in this Special Issue, and in other publications. We note that, given OMI's broad, worldwide user community, it is impossible to provide a complete overview all the achievements obtained using OMI data.

In addition to being a successful instrument, the OMI project has also brought together the research communities from the United States with the expertise from the Total Ozone Mapping Spectrometer (TOMS) (Heath et al., 1975; McPeters et al., 1998) and Solar Backscatter Ultraviolet (SBUV) (Cebula et al., 1988) instruments and European expertise based on the Global Ozone Monitoring Experiment (GOME) (Burrows et al., 1999) and the SCanning Imaging Absorption spectroMeter for Atmospheric CHartographY (SCIAMACHY) (Bovensmann et al., 1999; Noel et al., 2003).



This paper is organized as follows. In the introduction we present the OMI science questions, the data products, in-flight performance, and the instrument design features. The sections after the introduction cover the following themes:

- air quality monitoring, air quality forecasting, pollution events, and trends
- top-down emission estimates,
- monitoring of volcanoes,
- monitoring of the spectral solar irradiance,
- Montreal Protocal, total ozone, and UV-radiation,
- tropospheric ozone,
- research data products,
- multi-platform products and analyses,
- aircraft and field campaigns.

## 1.1 Objectives met and exceeded

At the start of the OMI project, the following four science questions were defined:

1. Is the ozone layer recovering as expected?
2. What are the sources of aerosols and trace gases that affect global air quality and how are they transported?
3. What are the roles of tropospheric ozone and aerosols in climate change?
4. What are the causes of surface UV-B change?

The first question was the main objective at the start of the OMI project. The OMI instrument has turned out to be very stable and provides a long data record for monitoring the ozone layer, which is critical for the assessment of the Montreal Protocol. As will be discussed in Sect. 6, the OMI data record covers a period where the ozone depletion has stopped and where we probably observe the onset of recovery.

The second science question deals with air quality where OMI clearly exceeded the expectations. By its frequent observations of trace gases such as nitrogen dioxide ($NO_2$), sulfur dioxide ($SO_2$) and formaldehyde, OMI contributed on research regarding the mapping of sources and transport of pollution, inversion modeling of emissions, as well as linking trends in air quality to policy measures. The OMI data show a steady decline in concentrations of $NO_2$ in the United States, Europe and Japan, whereas in China first strong increases were observed, followed by decreases after 2014 (van der A et al., 2017; Liu et al., 2016). These improvements can all be linked to the success of policy measures.

The third science objective considers the contribution of OMI to climate research by observing tropospheric ozone—a greenhouse gas and aerosols, which mainly act as cooling agents, although OMI is best at detecting absorbing aerosol that can cause warming. Tropospheric ozone can be derived from the OMI data itself (Sect. 7), or in combination with the Microwave Limb Sounder (MLS) and Tropospheric Emission Spectrometer (TES) instruments that are also manifested on the Aura platform as well as the Atmospheric Infra-Red Sounder (AIRS) on the EOS Aqua satellite that flies in formation with Aura. In both methods, it is important that a long-term data record on tropospheric ozone has been established. For aerosols, the focus has been on the absorption that can be derived in the UV. In combination with the TOMS, GOME and SCIAMACHY data, this is one of the longest aerosol data records available. Linked to aerosols are also the observations of $SO_2$, an important precursor for aerosol particles. The observations show that in the many parts of the world $SO_2$ is decreasing. However, in India we still observe strong increases due to the growing economy and the limited emissions control measures. Natural emissions of $SO_2$ by volcanoes have also been monitored by OMI in great detail (Sect. 4).

The last science question on the surface UV-B change is strongly linked to the long-term total ozone record. Research has focused on cases of high UV doses due to low total ozone (de Laat et al., 2010), showing a link in spring-time polar



ozone loss with UV-B in the following summer in the extratropics (Karpechko et al., 2013) and on explaining the differences between UV dose derived from satellite and measured on ground (Bernhard et al., 2015).

Whereas OMI was conceived as a research instrument, it also contributes to several operational applications. These applications make use of two data streams: the near real-time (NRT) data available within 3 hours of sensing and very-fast-delivery (VFD) data available within 20 minutes of sensing via the direct readout capability. Although these data streams were experimental, they turned out to be very successful. Operational users include the European Centre for Medium-range Weather Forecasts (ECMWF) and the US National Oceanic and Atmospheric Administration (NOAA) for ozone and air quality forecasts and the Volcanic Ash Advisory Centers (VAACs) for the rerouting of aircraft in case of a volcanic eruption. The NRT data are provided on the Tropospheric Emission Monitoring Internet Service (TEMIS) website (www.temis.nl) for the scientific user community. The VFD images are distributed at SAMPO website (sampo.fmi.fi).

### 1.2 Design features

In this section we describe some of the important new design features of OMI; for a detailed description of the instrument and its in-flight performance, we refer to Levelt et al. (2006), Dobber et al. (2006), van den Oord et al. (2006), and Schenkeveld et al. (2017) (this issue). OMI combines the spectral capabilities of the previously launched European spectrometers: the Global Ozone Monitoring Experiment (GOME) (Burrows et al., 1999) and SCanning Imaging Absorption SpectroMeter for Atmospheric CHartographY (SCIAMACHY) (Bovensmann et al., 1999; Noel et al., 2003) with the daily global mapping capability of NASA's Solar Backscatter UltraViolet (SBUV) and Total Ozone Mapping Spectrometer (TOMS) (Heath et al., 1975; Cebula et al, 1988; McPeters et al., 1998).

OMI is a nadir-looking, push broom UV/VIS solar backscatter grating spectrometer that measures the Earth radiance spectrum from 270-500 nm with a resolution of approximately 0.5 nm (Levelt et al., 2006a). The 114° viewing angle of the telescope corresponds to a 2600 km wide swath on the Earth's surface that enables measurements with a daily global coverage. The light entering the telescope is depolarized using a scrambler and then split into two channels: the UV channel (wavelength range 270 - 380 nm) and the VIS channel (350 - 500 nm). In the normal global operation mode, the OMI pixel size is $13 \times 24$ km$^2$ at nadir (along x across track). In addition to observing the Earth, OMI measures the solar irradiance once per day through the solar port.

OMI uses 2-dimensional (2D) detectors, where on one axis of the detector the across-track ground pixels are imaged and on the other axis the spectral information is recorded. This sensing technique allows the simultaneous measurement of all the ground pixels in the swath; therefore, OMI doesn't have a scan mirror. The 2D detectors enable the combinations of a wide swath, a good spatial resolution, and a high signal-to-noise ratio. The fact that most successor instruments, including the Ozone Mapping Profiler Suite (OMPS) nadir mapper launched in 2012 on the NASA/NOAA NPP satellite (Flynn et al., 2014), the TROPOspheric Monitoring Instrument (TROPOMI) to be launched in 2017 on the European Space Agency (ESA) Sentinel-5 Precursor (Veefkind et al., 2012), and the Environment Monitoring Instrument (EMI) to be launched in 2017 on the Chinese GaoFen-5 satellite (China Space Flight, 2017) are using this 2D imaging technique demonstrates the success of OMI. This technique is also used in the geostationary instruments that are currently in development, including the ESA Sentinel-4 (Ingmann et al., 2012), the Korean Geostationary Environmental Monitoring Spectrometer (GEMS) (Kim et al., 2012), and the NASA Tropospheric Emissions: Monitoring of Pollution (TEMPO) (Zoogman et al., 2016) mission. The high spatial resolution OMI (13 x 24 km$^2$ at nadir), was one of the key technical achievements that enabled significant advances in air quality research and emission monitoring from space and what motivates future air quality missions like TROPOMI to strive for even higher spatial resolution.

In addition to the 2D imaging technique, a new feature of OMI compared to GOME and SCIAMACHY, OMI incorporated the use of a polarization scrambler in a grating spectrometer. The polarization scrambler is applied in the OMI telescope before entering the polarization sensitive spectrograph; this makes the instrument almost insensitive to the



polarization of the incoming light. In GOME and SCIAMACHY, the polarization sensitivity is dealt with by measuring the degree of polarization at several wavelengths and applying complex correction algorithms. The OMI approach does not require such corrections, which simplifies the retrieval algorithms. One drawback is that a polarization scrambler can produce small spectral features that affect retrievals potentially leading to errors in trace gas concentrations. In addition, the

use of a polarization scrambler causes uncertainties in the spatial registration of the ground pixels. During the design phase, a careful trade-off between the amount of polarization scrambling, the spectral features, and the spatial registration was made. For the in-flight OMI data, these aspects have never been an issue. In many of the follow-on instruments polarization scramblers are also applied.

Another special feature of the OMI instrument is the type of diffuser used to observe the Sun. Such diffusers are required

to reduce the intensity of the solar radiance. In previous instruments, diffusers were made of aluminum or spectralon. These materials are optically stable but exhibit spectral features that can interfere with trace gas absorptions and thus affect the data quality. OMI has three diffusers that are used separately. Two of them are made of aluminum, whereas one is a new type Quartz Volume Diffuser (QVD). The new QVD had an unknown stability but was known to have smaller spectral features as compared with the aluminum diffusers. Because of its superior spectral behavior, the daily solar observations of OMI use the

QVD diffusor that has proven to be very stable. Therefore, the TROPOMI instrument only uses QVD solar diffusers.

### 1.3 In-flight performance

The in-flight performance is discussed in detail in Schenkeveld et al. (2017). Key aspects of the in-flight performance are the radiometric and spectral stability, the row anomaly, and detector degradation. The radiometric degradation of OMI ranges from ~2% in the UV channel to ~0.5% in the VIS channel, which is much lower than any other instrument of its kind.

Whereas it is not possible to reconstruct the causes of the low degradation, we believe that several aspects are important: the cleanliness during the instrument build and integration, the extensive outgassing period after the launch, and the fact that the primary mirror is protected in the instrument housing, instead of located outside the instrument housing which is needed in the case of a scanning mirror.

The one major anomaly of OMI is the so-called row-anomaly (Schenkeveld et al., 2017). A row anomaly is an anomaly

that affects the quality of the radiance data at all wavelengths for a particular viewing direction of OMI. This corresponds to a row on the 2D detectors, and hence the term 'row anomaly'. The cause for the row anomaly is outside of the instrument; it is most likely caused by damage to the insolation blankets in which OMI is covered, blocking part of the field of view. Although early signs are observed starting in 2007, the main row-anomaly started in 2009. For TROPOMI, the lesson learned was to put an additional aluminum plate over the insolation blankets at the location where the field-of-view is close

to the housing of the instrument.

The OMI 2D CCDs (Charge Coupled Devices) show several effects due to their exposure to the harsh space environment. First of all, the dark current has increased substantially. Overall, this degradation is as expected and can be corrected by subtracting daily measured dark current maps. However, some of the detector pixels show erratic changes in dark current over time, which is referred to as random telegraph signals (RTS). When the dark current of these pixels

changes significantly on time scales less than one day, they cannot be used in retrievals. This effect was known before the launch of OMI from the Global Ozone Monitoring by Occultation of Stars (GOMOS) (Betraux, et al., 2010) instrument on Envisat. At a late stage of the OMI development the detector temperatures were lowered to -8 °C and additional shielding was applied. It is recommended to further reduce the detector temperature to prevent significant effects of RTS. This is especially important of for trace gas retrievals with small absorption features such as for example formaldehyde.



### 1.4 OMI data products

In Table 1, a list is given of the standard, near real-time (NRT) and very fast delivery (VFD) products. The standard products are available within two days after measurement. OMI also provides global NRT data for selected products that are available within three hours after measurement. The VFD products are available for a limited region[1] covering most of

Europe twenty minutes after measurement. The algorithms used to generate the standard and NRT products are all published in peer reviewed papers, see table for references.

For some OMI level 2 (L2) data products, two standard algorithms have been developed. For example, for ozone there is a differential optical absorption spectroscopy (DOAS) retrieval and a TOMS-type of retrieval. This was considered important to be able to extend the existing DOAS data records from GOME and SCIAMACHY as well as the TOMS data

record. At the time OMI was launched, several $NO_2$ retrieval approaches were in development. For OMI, we developed an off-line $NO_2$ algorithm and an algorithm that runs in NRT to support air-quality forecasting applications. OMI does not have a separate cloud channel, like the $O_2$ A band, used by GOME and SCIAMACHY. Therefore, parallel development was started to ensure that at least one cloud data product would be ready. This resulted in two cloud products, one based on the $O_2$-$O_2$ absorption band and the other on rotational-Raman scattering. Both turned out to be successful and yield

complementary information. Although dual algorithm development might appear counterproductive, or seem confusing because users might not know which product to use, the experience within the OMI community has been of great benefit. A huge advantage of the development of two algorithms, using different physical approaches but the same OMI level 1B (L1B) dataset, is the added possibility of verification of the accuracy of the algorithm used and errors related solely to the retrieval technique and not the instrument errors. Additionally, two algorithms can be viewed as a type of ensemble result, a technique

widely used in the modelling community to get information on consistency between different modeling forecasts, and in turn the quality of the model forecast. This type of comparison has led to considerable improvements of both algorithms. In recent years, several OMI research data products have been developed, as well as some combined satellite data products (see Sect. 7, 8, and 9).

**Table 1: OMI standard products along with their type (L1B: radiances and irradiances, L2: Orbital data, L3: Gridded data) delivery method (S: Standard, NRT, or VFD), and PI organization (the Royal Netherlands Meteorological Institute, KNMI, the Finnish Meteorological Institute (FMI), National Aeronautics and Space Administration (NASA) and Smithosonian Astrophyical Observatory (SAO)).**

| Product Name | Product Type | Delivery Method | Principal Investigator Institute |
|---|---|---|---|
| Radiances and solar irradiances (OML1BRUG, OML1BRVG, OML1BRR) | L1B | S, NRT | KNMI, NASA |
| Aerosol absorption optical thickness and type (VIS) (OMAERO) | L2, L3 | S | KNMI |
| Aerosol absorption optical depth, and single scattering albedo (UV) (OMAERUV) | L2, L3 | S | NASA |
| BrO columns (OMBRO) | L2 | S | SAO |
| OClO slant column (OMCLO) | L2 | S | SAO |
| Cloud Product O2-O2 absorption (OMCLDO2) | L2 | S, NRT | KNMI |
| Cloud Product Rotational Raman (OMCLDRR) | L2 | S, NRT | NASA |

---

[1] VFD products cover roughly an area northwards from northern Italy and Spain and from Greenland in west to Ural mountains in east.



| HCHO columns (OMHCHO) | L2 | S | SAO |
|---|---|---|---|
| NO$_2$ column (standard) (OMNO2) | L2, L3 | S | NASA, KNMI |
| NO$_2$ columns (DOMINO) | L2, L3 | NRT | KNMI |
| O$_3$ total column, aerosol index[2] (TOMS) (OMTO3) | L2, L3 | S, NRT | NASA |
| O$_3$ total column (DOAS) (OMDOAO3) | L2, L3 | S, NRT | KNMI |
| O$_3$ profile (OMO3PR) | L2 | S | KNMI |
| Pixel Corners (OMPIXCOR) | L2 | S | NASA |
| SO$_2$ columns (OMSO2) | L2, L3 | S, NRT | NASA |
| OMI MODIS Merged Cloud (OMMYDCLD) | L2 | S | NASA |
| OMI Indices collocated to MODIS Aerosol products (OMMYDAGEO) | L2 | S | NASA |
| Surface Reflectance Climatology (OMLER) | L3 | S | KNMI |
| Surface UVB (OMUVB) | L2, L3 | S | FMI |
| Total O$_3$ | L2 | VFD | FMI, KNMI, NASA |
| Effective cloud fraction | L2 | VFD | FMI, KNMI |
| UV index, Erythemal daily UV dose | L2 | VFD | FMI |
| SO$_2$ columns | L2 | VFD | FMI, NASA |
| Aerosol index | L2 | VFD | FMI, KNMI, NASA |

**2 Air quality monitoring, air quality forecasting, pollution events and trends**

OMI collects information on several key pollutants including aerosols, O$_3$ (discussed in Sect. 7), nitrogen dioxide (NO$_2$) (See Fig. 1), sulfur dioxide (SO$_2$), and formaldehyde (HCHO, an air toxin), all of which contribute to morbidity and

5   mortality (WHO, 2014). Air pollution causes 1 in 9 deaths globally (WHO, 2016), costing the global economy $225 billion in lost labor income annually and more than $5 trillion in welfare losses (World Bank, 2016). By 2060, 6 to 9 million annual premature deaths are expected with annual global welfare costs projected to rise to US $18-25 trillion. Ecosystem health is also degraded by air pollution, such as by acid rain, eutrophication of water bodies, and oxidation of plant tissue by ozone (O$_3$). Reduced global crop yields are estimated at about 10% annually (Van Dingenen et al., 2009; Fishman et al., 2010;

10   Avnery et al., 2011), with some heavily polluted areas, like parts of India, experiencing a 50% reduction (Burney and Ramanathan, 2014).

---

[2] The aerosol index is currently part of the OMTO3 product, but will be transition soon to the OMAERUV product.




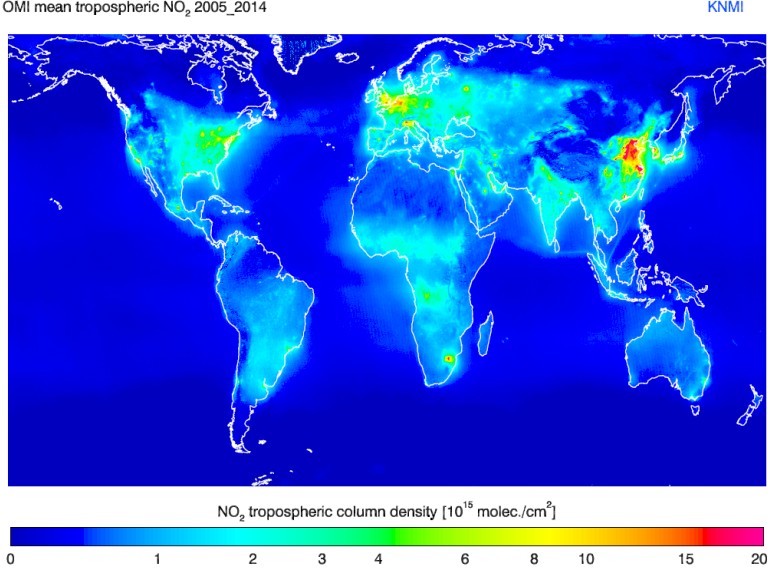

**Figure 1: The mean tropospheric NO2 concentrations based on 10 years (2005-2014) of OMI observations. Note that an exponential scale is used to see more details.**

5      OMI's spatial coverage is far greater than can be provided by surface monitoring networks, which, for example, increases statistical power that strengthens inference of the relation between pollutants and health outcomes. For example, Clark et al. (2014) use OMI $NO_2$ data to estimate that reducing U.S. nonwhites' exposure to $NO_2$ concentrations to levels experienced by whites would reduce coronary heart disease mortality by about 7,000 deaths/year.  They argue that their results may aid policy-makers in identifying locations with high environmental injustice and inequality. The defining

10    strength of OMI is that it currently provides the finest spatial resolution as compared with previous and present instruments that make measurements in the same spectral range.  As an example, Fig. 2 illustrates the comprehensive global coverage of OMI $NO_2$ data, but also the unprecedented detail in air pollution changes, down to sub-urban scales (Duncan et al., 2016). Consequently, OMI data are being increasingly exploited for a wide variety of air quality and health applications and in decision making activities (e.g., Streets et al., 2013; Duncan et al., 2014).

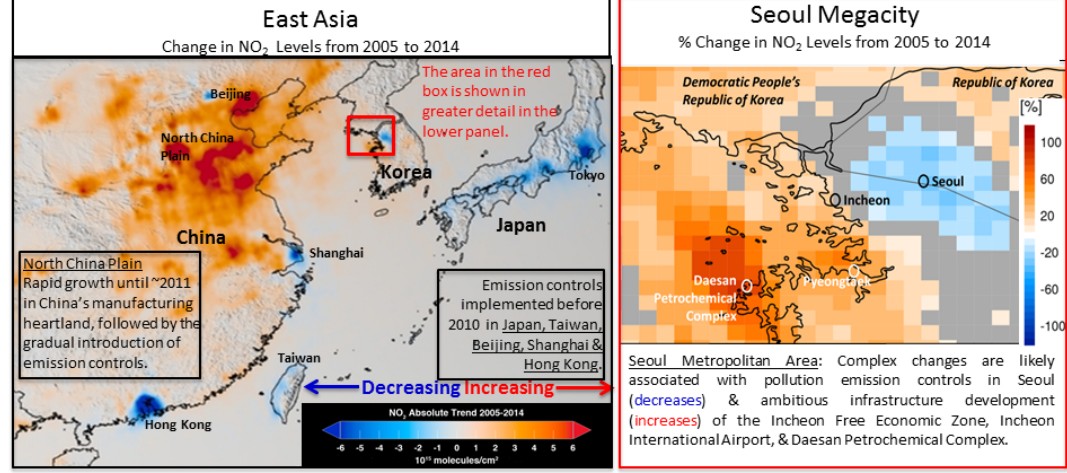



**Figure 2: Left: The world, particularly East Asia, shows intriguing spatial heterogeneity and changes in NO$_2$ air pollution from 2005 to 2014 (Duncan et al., 2016). Right: Ongoing retrieval algorithm work allows for unprecedented detail, including sub-urban scales, such as in Seoul.**

Continuing refinements to the OMI retrieval algorithms have resulted in data products that are of sufficient maturity to allow for the reliable and quantitative estimation of concentrations, trends, and fluxes of surface pollutants. However, there are remaining challenges (Martin 2008; Streets et al. 2013; Duncan et al. 2014; and references therein) that are being addressed. For instance, a fundamental challenge of using these data is the proper "translation" of the observed quantities to more useful surface quantities (Lamsal et al., 2008), such as emissions and concentrations. From the OMI spectra, one infers

a column density, which is typically reported in units of molecules/cm$^2$. From a column density, one may infer a surface concentration or emission flux if the majority of the temporal variation within the column density is associated with near-surface sources. This is the case for NO$_2$, SO$_2$, and formaldehyde as their chemical lifetimes are short and their primary sources are located near the Earth's surface. As an example, Lamsal et al. (2015) show that the long-term trends and short-term monthly variations of OMI NO$_2$ column densities from 2005 to 2013 compare well (e.g., r = 0.68 for trends) with those

from the majority of surface concentration observations from the U.S. Environmental Protection Agency's (EPA) Air Quality System (AQS). Lamsal et al. (2015) argue that the spatial coverage afforded by the OMI satellite data in combination with the maturity of the current retrieval algorithm allows for a more representative estimation of NO$_2$ trends within a city than the observations from the sparse surface monitors, which typically number < 5 in most major U.S. cities.

### 2.1 Applications of OMI Data for Health and AQ Studies

The use of OMI data by the health and air quality communities has grown dramatically within the last few years. For instance, OMI total ozone column data have been used in several studies to understand the impact of UV exposure on human health (e.g., Beckett et al., 2016; Lucock et al., 2016) similar to the earlier TOMS records (e.g., Boscoe et al., 2006; Chang et al., 2010). A consistent long-term global satellite UV radiation time series is useful for several health related studies (Langston et al., 2017). For HCHO and NO$_2$, the use of the data for health studies has definitely benefited from recent

advances in data quality that are the direct result of ongoing OMI retrieval algorithm improvement (Boersma et al., 2011; Bucsela et al., 2013; González Abad, 2015; van Geffen et al., 2015; Marchenko et al., 2015; Krotkov et al., 2017).

### 2.1.1 NO$_2$

OMI NO$_2$ data have been used in a number of recent health studies (e.g., Hystad et al., 2011, 2012; Novotny et al., 2011; Prud'homme et al., 2013; Vienneau et al., 2013; Knibbs et al., 2014; Hoek et al., 2015; Belche et al., 2015; Crouse et al.,

2015; de Hoogh et al., 2016; Young et al., 2016). For example, Belche et al. (2013) found that annual OMI NO$_2$ column density data correlate well (r=0.93) with surface data in southern California and provide a reliable measure of spatial variability for NO$_2$ exposure assessment. NO$_2$ has adverse health effects and is correlated with morbidity and mortality (Brook et al., 2007; WHO, 2014), though this correlation may occur because many short-lived air toxics are co-emitted with NO$_2$ and it is a key player in the formation of unhealthy levels of surface ozone (Brook et al., 2007). In fact, Brook et al.

(2007) concluded that NO$_2$ is a better indicator than PM$_{2.5}$ of a range of pollutants (e.g., Volatile Organic Compounds (VOCs), aldehydes, oxidized nitrogen species, and particle-bound organics) from vehicle exhaust. The use of OMI NO$_2$ data for health studies is attractive given recent advances in the quality of the data that has improved agreement between the data and independent quantities, such as surface NO$_2$ levels and NO$_x$ emissions (e.g., Boersma et al., 2009; Knepp et al., 2013; Lamsal et al., 2015; Duncan et al., 2013, 2016), and improved techniques to infer surface concentrations from satellite data

(Lamsal et al., 2008). For example, Ialongo et al. (2016) compared the weekly and seasonal cycle in satellite-based NO$_2$ data and surface concentrations from an air quality station at a high-latitude urban site (i.e, Helsinki, Finland). Despite the



challenging viewing conditions and frequent cloud-contamination, OMI NO$_2$ observations have shown their capability to describe air quality features also at relatively high latitudes.

### 2.1.2 HCHO

HCHO is an important VOC, acts as an ozone precursor, is associated with the formation of organic aerosols, and is an important carcinogen in outdoor air. It is produced from oxidation of methane and isoprene, and is thus strongly linked to natural emissions. There are also important anthropogenic emissions associated with a range of industrial activities, mostly in the oil- and gas refining sectors (e.g., Zhu et al., 2014). Trend studies with OMI formaldehyde retrievals indicate increases in HCHO columns over India and China, and a downward trend over the Amazonian forest, spatially correlated with areas affected by deforestation (De Smedt et al., 2015).

OMI HCHO data have been used in a number of studies, including to infer health outcomes and to provide top-down constraints on the emissions of VOCs. For example, Zhu et al. (2017) use OMI HCHO data to estimate that 6,600-12,200 people in the U.S. will develop cancer over their lifetimes by exposure to outdoor HCHO derived from biogenic VOC emissions. However, the HCHO yield from VOC oxidation is proportional to NO$_x$ level, so Zhu et al. note that NO$_x$ emission controls to reduce O$_3$ have the co-benefit of reducing HCHO-related cancer risks. Marais et al. (2012) used OMI

HCHO data to infer isoprene emissions from Africa and suggested that the MEGAN inventory may significantly overestimate emissions for the region. Millet et al. (2008) found a similar positive bias in MEGAN for North America as compared with OMI retrievals. Zhu et al. (2014) applied the over sampling technique to OMI HCHO data, and suggested that anthropogenic emissions of highly reactive VOCs from the Houston area could be several times larger than the EPA estimates. These efforts are currently limited by relatively large uncertainties in satellite retrievals of HCHO, as

demonstrated by significant differences in top-down estimates using different sensors (e.g., Barkley et al., 2013). More validation efforts (e.g., Zhu et al., 2016) may help to improve the quality of OMI HCHO data in the future.

### 2.1.3 HCHO and NO$_2$ as O$_3$ precursors

OMI NO$_2$ and HCHO data serve as effective proxies for NO$_x$ (= NO + NO$_2$) and volatile organic compounds (VOCs), respectively, both necessary ingredients for the formation of unhealthy levels of surface ozone. Martin et al. (2004)

demonstrated that the ratio of HCHO to NO$_2$ column densities can be an effective indicator of ozone's production sensitivity to NO$_x$ and VOC emission reductions. This information is important for the development of effective ozone pollution mitigation strategies. Duncan et al. (2010) used the ratio of OMI HCHO to NO$_2$ columns to show that the chemical sensitivity of ozone formation was becoming more sensitive to NO$_x$ levels in U.S. cities, even cities that were typically considered to be more sensitive to VOCs (e.g., Los Angeles), as a result of substantial reductions in NO$_x$ emissions. Over

China, Jin and Holloway (2015) found a complex result owing to significant spatial heterogeneity of NO$_x$ and VOC emission changes during the OMI record.

### 2.2 Improving Models and AQ Forecasting

OMI data are being used to improve AQ forecasting in a number of ways. For instance, OMI NO$_2$ data have been used in several recent studies to identify likely inaccuracies in the chemistry, dynamics, and emissions of AQ models. Travis et al.

(2016) used a combination of NASA OMI NO$_2$ data, NASA SEAC$^4$RS field campaign data, and an atmospheric model to show that industrial and mobile source NO$_x$ emissions in the U.S. EPA National Emission Inventory (NEI) are likely 30-60% too high. This finding has broad implications for identifying (with AQ models) the most effective and cost-effective strategies to improve AQ. Based on their evidence, Travis et al. (2016) adjusted the NEI NO$_x$ emissions in their atmospheric model, which reduced part of the high bias in simulated O$_3$. A high bias in simulated O$_3$ relative to observations has been a

chronic problem of all atmospheric models over the eastern U.S. Canty et al. (2015) also used OMI data to diagnose a likely



high bias in NEI NO$_x$ emissions, but also in the chemical representation of alky nitrates in a chemical mechanism of an AQ model.

The Atmosphere Monitoring Service of the European Copernicus Programme (CAMS, atmosphere.copernicus.eu) is an operational service providing validated (Eskes et al., 2015) analyses, reanalyses, and daily forecasts of aerosols, reactive

gases, and greenhouse gases on a global scale, and AQ forecasts and reanalyses on a regional scale (Marécal et al., 2015). In CAMS, data assimilation techniques are applied to combine in-situ and remote sensing observations with global and European-scale models of atmospheric reactive gases, aerosols, and greenhouse gases. The global component is based on the Integrated Forecast System of the ECMWF, and the regional component on an ensemble of seven European air quality models. OMI, and in the near future the Sentinel 5P TROPOMI instrument (Veefkind et al., 2012), are providing crucial

datasets for the CAMS assimilation system. OMI data are extensively used in both the global and regional components of CAMS. In the global component, OMI observations of the total ozone column have been assimilated from September 2009 onwards, and OMI NO$_2$ and SO$_2$ are assimilated since July 2012 (Inness et al., 2015). OMI measurements have been used in the last reanalysis (2003-2012 period) produced in the MACC project, the precursor of CAMS (Inness et al., 2013). For the regional air quality service, all 7 models have developed data assimilation capabilities for daily air pollution analyses and

yearly reanalyses for Europe (Marécal et al., 2015). Here the prime focus is the assimilation of surface observations from the European regulatory network, but several regional models have included OMI NO$_2$. Figure 3 shows results of the assimilation of OMI NO$_2$ data in the Lotos-Euros regional air quality model which is one of the CAMS ensemble members. The LOTOS-EUROS model has also been used to study trends of NO$_2$ over Europe (Curier et al., 2014).

A limited number of regional initiatives use OMI satellite products in AQ forecast systems to provide timely AQ to

citizens. Over the U.S. Pacific Northwest, OMI NO$_2$ data has been used mostly to evaluate the air quality forecasting system (Herron-Thorpe et al., 2010). In France, a system was developed that assimilates OMI NO$_2$ with an optimal-interpolation method in an air quality model to improve NO$_2$ forecasts in Europe (Wang et al., 2011). The assimilation results in an improved capacity of the system to predict NO$_2$ pollution. A similar system was developed by Silver et al. (2013), who showed that the assimilation of OMI tropospheric NO$_2$ columns leads to an improved agreement between predicted and

observed surface NO$_2$ concentrations over Europe; they also noted that the effect of assimilation is fairly small and local.





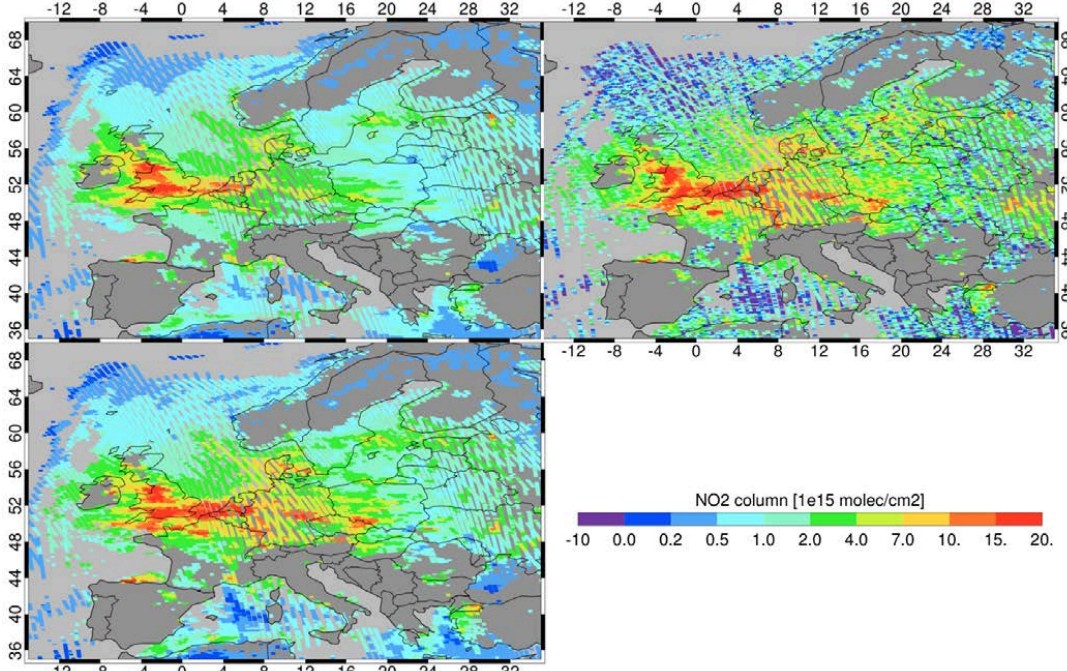

**Figure 3: Example result of the assimilation of OMI NO₂ tropospheric column amounts (DOMINO-2 product, upper right) on 26 March 2007 in the LOTOS-EUROS regional air quality model (model simulation for OMI ground pixels is provided in the upper left by spatial interpolation and by applying the averaging kernels). LOTOS-EUROS makes**
**use of the ensemble Kalman filter to optimise model processes by assimilating observations. In this case the surface NOₓ emissions are adjusted to optimise the match with the tropospheric NO₂ columns observed. The resulting analysis is shown in the lower-left panel. OMI observations for cloud-covered scenes (cloud radiance fraction > 50%) are removed.**

The spatial footprint of OMI aerosol optical depth (AOD) data is broader as compared to data from other instruments, though the OMI products include several important and some unique quantities that give important information on aerosol absorption properties of absorbing species such as dust and smoke. These OMI data products include near UV (OMAERUV) aerosol record of AOD, single scattering albedo (SSA) and aerosol index (AI) (Torres et al., 2007). Retrieved AOD and SSA products have been evaluated using ground-based observations (Ahn et al., 2014; Jethva et al., 2014a) as well

as other satellite based products (Ahn et al., 2008; Gasso and Torres, 2016). OMAERUV observations of aerosol UV absorption spectral dependence are being used to improve tropospheric photochemistry modeling capabilities. As shown by Hammer et al. (2016), when including brown carbon aerosol absorption in the simulation of OMI aerosol observations using an atmospheric model coupled with radiative transfer calculations, the observed discrepancies between simulated and observed hydroxyl radical (OH) concentrations are significantly reduced. The inclusion of brown carbon aerosol decreases

OH by up to 35% over South America in September, up to 25% over southern Africa in July, and up to 20% over other biomass burning regions. Modeled global annual mean OH concentrations decrease due to the presence of absorbing brown carbon aerosol, thus reducing the bias against observed values.

### 2.3 Pollutant Trends

    OMI's data have been used to infer substantial trends that have occurred in SO₂ and NO₂ pollution around the world over
the last decade (e.g., Duncan et al., 2016; Krotkov et al., 2016). These changes are largely consistent with the implementation of environmental regulations on emissions and changes in economic output, including changes resulting





from the global economic recession of 2008-2009 (e.g., Castellanos and Boersma, 2011; Russell et al. 2012; Boersma et al., 2015; Duncan et al., 2016, Krotkov et al., 2016). For example, Fioletov et al. (2011), Zhou et al. (2012), and Duncan et al. (2013) used OMI $SO_2$ and $NO_2$ column density data, respectively, to quantify the substantial reductions in pollution over U.S. and Spanish power plants, which primarily resulted from the implementation of emission control devices. They showed

that the changes in the OMI column densities agree well with changes in power plant emissions reported to the U.S. EPA Continuous Emissions Monitoring System (CEMS). Over the eastern US, both $NO_2$ and $SO_2$ levels decreased dramatically from 2005 to 2015, by more than 40 and 80%, respectively, as a result of both technological improvements and stricter regulations of emissions. Similarly, OMI confirmed large reductions in $SO_2$ over Eastern Europe's largest coal-fired power plants after installation of flue gas desulfurization devices. The North China Plain, China's manufacturing heartland, has the

world's most severe $SO_2$ pollution, but a decreasing trend has been observed since 2011, with about a 50 % reduction in 2012–2015, due to an economic slowdown and government efforts to restrain emissions from the power and industrial sectors. In contrast, India's $SO_2$ and $NO_2$ levels from coal power plants and smelters are growing at a fast pace, increasing by more than 100 and 50 %, respectively, from 2005 to 2015. In stark contrast to decreasing surface pollution in the United States and Europe, the booming Chinese and Indian economies and limited environmental regulation of emissions led to

large increases in $NO_2$ and $SO_2$ levels indicated by OMI (e.g., Wang et al. 2012; Li et al. 2010; Lin and McElroy, 2011; Lu and Streets, 2012; Verstraeten et al., 2015). However, OMI data show that recent Chinese emission control efforts have led to improvements in air quality (e.g., Krotkov et al., 2015; van der A et al., 2017). Recently, Boersma et al. (2015) used OMI $NO_2$ observations to derive the changes in polluting emissions from European shipping.

OMI data can also be used in combination with other satellite datasets to establish even longer data records useful for

trend analysis, as has been carried out for the aerosol index (Popp et al., 2016) and total ozone column (see Sect. 6.1). As can be seen in Fig. 4, the aerosol index covers a period of nearly 40 years and brings together data from both the European and American communities (TOMS (Nimbus 7), GOME, SCIAMACHY, OMI, GOME-2A, and GOME-2B). This data record can be used to better understand regional and global trends in the presence of UV-absorbing aerosols including desert dust and biomass burning aerosols.

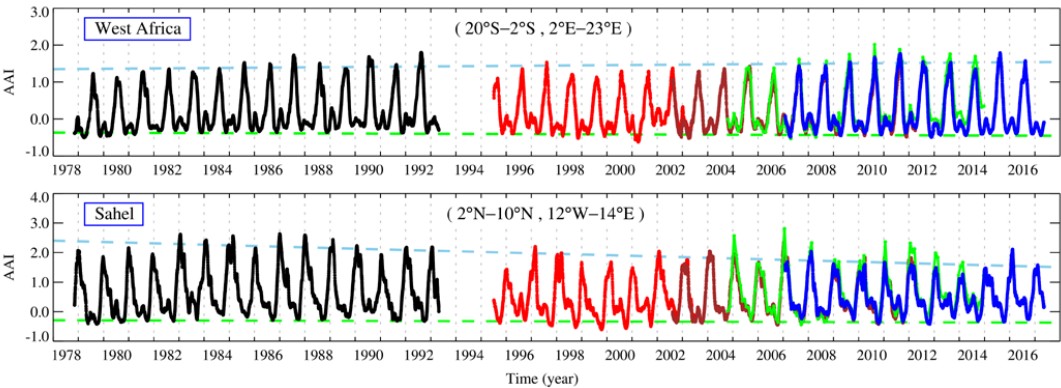

**Figure 4: Time series of regional mean AAI for two aerosol regions. The time series consists of data from TOMS (black), GOME (red), SCIAMACHY (brown), OMI (light green), and GOME-2 (blue). The latitude and longitude ranges that define the regions are provided in the plot windows. The dashed green lines indicate linear fits to the**
**yearly minima of the time series, and illustrate the stability of the data records. The dashed blue lines represent linear fits to the yearly maxima of the time series. These describe trends in aerosol presence for the regions over the entire time range covered by the time series.**




Over eastern Asia, Verstraeten et al. (2015) showed that strong increases between 2005-2010 in OMI $NO_2$ columns can been used to quantify the contribution of photochemical ozone formation to the rapid increase in mid-tropospheric ozone concentrations over and downwind of that continent. Their study demonstrated that the good vertical sensitivity of OMI to ozone precursors down to the Earth's surface can be combined with the sensitivity of the TES instrument to mid-

tropospheric ozone, to arrive at a more comprehensive understanding of spatio-temporal patterns in tropospheric ozone.

### 3 Top-down Emissions Estimates

OMI data have played a key role in the top-down estimation of $NO_x$, $SO_2$, and VOC emissions. Particulate matter (PM) emissions may be inferred via OMI AOD measurements, but a direct relationship with PM emissions is still elusive (e.g., Hoff and Christopher, 2009). Because statistical data needed by bottom-up inventories often take years to collect, the short-

term availability of satellite data is often used to show the latest trends in emissions and the effectiveness of air quality regulations (e.g., de Foy et al., 2016; Duncan et al., 2016; Krotkov et al., 2016). The high resolution OMI observations allow the emission sources to be resolved at a higher resolution than before, which is a distinct advantage for point sources of short-lived gases, including $NO_2$ and $SO_2$, since their sources can be derived with relatively simple methods based on mass balance (e.g., Duncan et al., 2013; de Foy et al., 2015; Fioletov et al., 2015, 2016; Liu et al., 2016; McLinden et al., 2016a).

Complete emission maps from OMI observations have been derived using full inversion methods that involves the use of chemical transport models e.g., Qu et al., 2017. Streets et al. (2013) reviewed the current capability to estimate emissions from space, and in this section we highlight studies of emissions using OMI data that have been published subsequently.

### 3.1 $NO_x$ emission estimates

The top-down estimation of $NO_x$ emission sources are especially successful because of the strength of the OMI signal

and therefore its potential to detect low-intensity sources. Applications have included ship emissions (Vinken et al. 2014a), Canadian oil sands (McLinden et al. 2014), soil emissions (Vinken et al. 2014b), biomass burning (Castellanos et al. 2014), and urban areas (Vienneau et al. 2013, Ghude et al., 2013). Another recent development has been the application of OMI $NO_x$ data to studies of nitrogen deposition flux (Nowlan et al. 2014). Trend studies of $NO_x$ point sources were performed by, for example, De Foy et al. (2015) who derived $NO_x$ emissions from 29 isolated power plants in the USA and Lu et al. (2015)

estimated summertime $NO_2$ emissions from 35 US urban areas. A slightly different method was used by (Liu et al., 2016) to estimate emissions and lifetimes for 17 power plants and 53 cities located in non-mountainous regions across China and the US.

Full inversion emission estimates for $NO_x$ from OMI observations using a chemical transport model (CTM) were continued to be improved by various research groups. Stavrakou et al. (2013) applied a 4DVAR inversion on OMI $NO_2$

observation using a global CTM. Miyazaki et al. (2012, 2013) applied an Ensemble Kalman Filter scheme on observations of multiple species of OMI. Mijling and van der A (2012) further developed their Daily Emission estimates Constrained by Satellite Observations (DECSO) algorithm for high resolution regional emission estimates using an Extended Kalman Filter. DECSO has been applied for the regions Europe, East China, India, South Africa and Middle East. Figure 5 shows NOx emissions in the Middle East based on the latest bottom-up inventory, the Emission Database for Global Atmospheric

Research (EDGAR) v4.3 and the DECSO algorithm version 3b. Because of the fast availability of satellite-derived emissions, the DECSO results show the change of ship routes along the coast of Somalia as a result of the increased number of piracy cases. Notably in the DECSO emission map is also the dispersion of ships East of Yemen and the more realistic stronger economic activity in the Persian Gulf region. With the improved DECSO algorithm version 5 even ship tracks hidden under the strong outflow of pollution along the Chinese coast became clearly visible (Ding et al., 2017a). The

possibilities of high resolution emissions is demonstrated by Ding et al. (2015) who showed the temporal effect of air quality



regulations on city emissions during the Youth Olympic Games in Nanjing. Trends in these $NO_x$ emissions based on 12 years of OMI observations have been analysed by Mijling et al. (2013) for China, by van der A et al. (2017) for China including the relation with air quality regulations, and by Miyazaki et al. (2017) on a global scale. Emission inventories over China were validated by a detailed intercomparison of five bottom-up inventories and four satellite-derived emission

inventories using GOME-2 and OMI (Ding et al., 2017b).

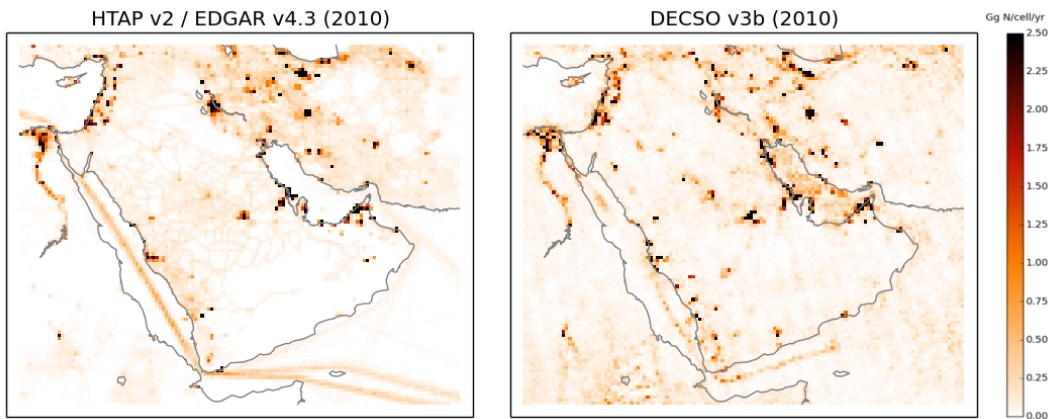

**Figure 5: $NO_x$ emissions in the Middle East in 2010 derived from (left) the bottom-up inventory EDGAR v4.3 and (right) the DECSO algorithm v3b applied to OMI $NO_2$ observations.**

### 3.2 $SO_2$ emission estimates

Though the $SO_2$ signal from OMI is two to three orders of magnitude weaker than the $NO_2$ signal, oversampling[3] and other data enhancement techniques have enabled valuable new studies of $SO_2$ emissions from Canadian oil sands (McLinden et al., 2014). Also, using specific sampling techniques, OMI observations were used as the first satellite observation of $SO_2$

ship track emissions (Theys et al., 2015). Fioletov et al. (2013) reviewed the ability of OMI to detect large $SO_2$ sources worldwide, including power plants, oil fields, metal smelters, and volcanoes. Work continues on the challenge of developing reliable quantitative relationships between OMI observations and emissions for large isolated sources. Previous work had only moderate success in correlating observations with emissions. An alternative approach, well-suited for deriving emissions from continuously emitting (near-) point sources and which does not require the use of atmospheric chemistry

models, is based on merging OMI measurements of tropospheric columns with wind information and examining the downwind decay of the pollutants (Beirle et al., 2011). This approach spawned several studies on $SO_2$ emissions (Fioletov et al., 2011, 2015; de Foy et al., 2015; Lu et al., 2013, 2015; Wang et al., 2015) utilizing increasingly complex analysis methods in which an estimate of the total mass near the source and its lifetime or, more accurately, decay time were derived. Assuming a steady state, the emission strength can be obtained from the ratio between mass and decay time. The mass can

be derived directly from satellite measurements, while the lifetime can either be prescribed using known emissions (Fioletov et al., 2013, 2016) or estimated from the measurements based on the rate of decay of VCD with distance downwind (Beirle et al., 2014; Carn et al., 2013; de Foy et al., 2015).

---

[3] The fact that daily pixels do not match spatially can be used to sample data to a grid that is smaller than the pixel size. This is called oversampling.



OMI SO$_2$ data, significantly improved with a principal component analysis algorithm (Li et al., 2013), was combined with a new emission-source detection algorithm (Fioletov et al., 2015) to compile the first global, satellite-based emissions inventory of point SO$_2$ emission sources (Fioletov et al., 2016). The inventory contains estimates of annual emissions for 491 medium to large sources (volcanoes, power plants, oil and gas related sources, and smelters) that emit from 30 kt y$^{-1}$ and is

completely independent of conventional information sources. It was used for verification of traditional "bottom-up" SO$_2$ emissions inventories and identification of missing sources. Nearly 40 of the sources identified by this new method were found to be missing from leading emissions inventories, representing about 12% of the global total (McLinden et al., 2016b). Regionally, emissions can be off by factors of 2 or 3. Many of the missing SO$_2$ sources were located in the Middle-East and related to the oil & gas sector (Fig. 6). OMI is also able to capture annual variability for SO2 emissions for all detected

sources of magnitude 30-4000 kt/yr when averaged over 2005-2015 (Fioletov et al., 2016) .

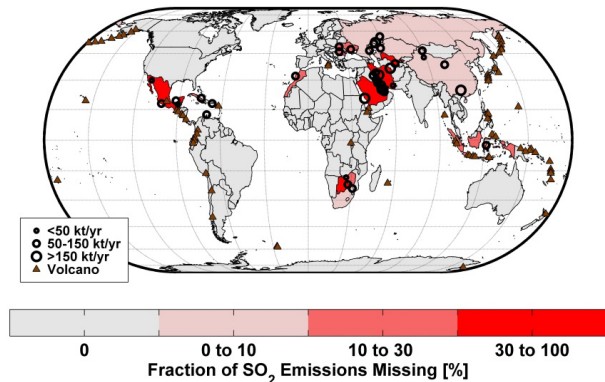

**Figure 6: Point SO$_2$ sources identified that were found to be missing from three leading bottom-up inventories 16–18 (from McLinden et al., 2016b). Each nation is color-coded according to its total fraction of SO$_2$ emissions that are**
**missing, relative to its total national emissions (the sum of HTAP and missing).**

**3.3 VOC emission estimates**

OMI detects the small organic molecules formaldehyde and glyoxal. OMI formaldehyde data have been used to infer natural emissions of isoprene, a key contributor to O$_3$ production in many parts of the world and the largest VOC source
globally (e.g., Millet et al., 2008; Duncan et al. 2009, 2010; Curci et al., 2010; Marais et al. 2012, 2014; Barkley et al., 2013; Zhu et al. 2014; Stavrakou et al., 2015; Bauwens et al., 2016). More recently, Valin et al. (2016) investigated the influence of the hydroxyl radical (OH) and VOC variability on the OMI HCHO column, which is important when inferring fluxes of VOCs using OMI HCHO columns. They conclude that the column primarily depends on OH production rates (POH) at low OH concentrations, on both POH and VOC reactivity (VOCR) at moderate OH, and on VOCR at high OH. OMI
observations have also been used to estimate anthropogenic emissions of highly reactive VOCs over the Southeast USA (Zhu et al., 2014). VOC emissions derived from OMI formaldehyde observations showed that crop burning in the China North Plain was underestimated with a factor two in traditional emission inventories (Stavrakou et al., 2016). The combination of OMI HCHO and OMI glyoxal measurements (see Sect. 8.3) can provide useful information on VOC speciation (Di Gangi, 2012; Chan Miller et al., 2016) and can constrain VOC emissions (Stavrakou et al., 2009).



### 3.4 NO$_2$ as indicator for anthropgenic CO$_2$ emissions

OMI observations are being synergistically combined with observations from other satellite instruments to provide information, such as the quantification of emissions, inference of co-emitted trace gases, and the identification of source regions and types, that neither instrument could do alone. These recent developments in the combined interpretation of NO$_2$ and CO$_2$ satellite emissions have gained much scientific attention. Duncan et al. (2016) showed that OMI NO$_2$ can differentiate individual power plant sources even in complex source regions and proposed that NO$_2$ may be used to infer CO$_2$ emissions assuming a characteristic CO$_2$/NO$_2$ emission ratio. Hakkarainen et al. (2016) show that OMI NO$_2$, an indicator of atmospheric pollution, may be used to aid in the interpretation of the NASA Orbiting Carbon Observatory-2 (OCO-2) carbon dioxide (CO$_2$) data. The spatial distribution of OMI tropospheric NO$_2$ matched the features observed in the maps of OCO-2 CO$_2$ anomalies over the main polluted regions. Furthermore, the results of a cluster analysis between OMI NO$_2$ and OCO-2 CO$_2$ confirmed the spatial correlation over areas with different amounts of pollution. Konovalov et al. (2016) found that OMI NO$_2$ data can provide a better constraint than IASI CO data for anthropogenic CO$_2$ emissions. This is important as inferring emissions with current satellite CO$_2$ datasets (e.g., OCO-2, GOSAT) is challenging for a variety of reasons.

### 4 Volcanic monitoring with OMI

OMI can lay claim to being the first satellite instrument to be used for daily monitoring of volcanic emissions (e.g., Carn et al., 2008; Carn et al., 2013; McCormick et al., 2013; Flower and Carn, 2015), heralding a new era where satellite measurements have become an indispensable tool for volcanic gas monitoring in many regions. While instruments such as TOMS have been measuring SO$_2$ and ash emissions by major eruptions since 1978 (e.g., Krueger, 1983; Carn et al., 2016), and GOME first demonstrated the potential for detection of tropospheric volcanic SO$_2$ from space by hyperspectral UV sensors (Eisinger and Burrows, 1998), the 'volcano-scale' pixel size (13×24 km at nadir) of OMI was a critical factor. OMI's ability to detect volcanic SO$_2$ at all levels from the planetary boundary layer (PBL) to the stratosphere, derived from volcanic activity of varying intensity from passive degassing to major stratospheric eruptions, has required the development of SO$_2$ retrieval algorithms capable of spanning several orders of magnitude of SO$_2$ column amount (e.g., from 0.2-2000 DU; Krotkov et al., 2006; Yang et al., 2007, 2009b, 2010; Li et al., 2013; Theys et al., 2014; Li et al., 2016) and direct retrieval of SO$_2$ altitude from UV radiances (e.g., Yang et al., 2009a, 2010).

Unlike the 1978 to 2005 period of TOMS measurements, which featured the major SO$_2$-rich eruptions of El Chichón (Mexico) in 1982 (Krueger, 1983) and Pinatubo (Philippines) in 1991 (Bluth et al., 1992), the decade since the OMI launch has seen no eruptions of comparable magnitude (Carn et al., 2016). The largest tropical eruption occurred at Nabro (Eritrea) in June 2011, but was an order of magnitude smaller than Pinatubo (Goitom et al., 2015). Nevertheless, the OMI era has been notable for a number of large, high-latitude eruptions (e.g., 2008 Okmok, 2008 Kasatochi, 2009 Sarychev Peak, 2009 Redoubt, 2014 Holuhraun), with the eruption of Kasatochi (Aleutian Islands, USA) in August 2008 representing one of the largest stratospheric SO$_2$ injections of the last decade (e.g., Krotkov et al., 2010; Wang et al., 2013). Although none of these eruptions were large enough to impact climate (due to the high latitude and insufficient SO$_2$ release), they have presented several opportunities for OMI SO$_2$ validation, owing to the limited latitudinal spread of the volcanic clouds and the abundance of ground stations at mid- to high-latitudes (e.g., Spinei et al., 2010; Carn and Lopez, 2011; Lopez et al., 2013; Ialongo et al., 2015). Operation of Aura in the A-Train constellation has facilitated validation by providing critical observations of volcanic cloud altitude (e.g., from CALIPSO; Spinei et al., 2010; Carn and Lopez, 2011).

Despite the lack of major, climate-forcing eruptions, the OMI era has been far from volcanically quiescent (e.g., Prata et al., 2010; Carn and Prata, 2010; Carn et al., 2016). The absence of significant volcanic perturbations to stratospheric AOD in the post-Pinatubo period to date has focused attention on the impacts of smaller, more frequent volcanic eruptions on



'background' stratospheric AOD and related subtle climate impacts (e.g., Solomon et al., 2011; Vernier et al., 2011; Santer et al., 2014). Although the implication of increased rates of volcanic activity in 2000-2010 in a putative 'global warming hiatus' is now deemed unlikely, accurate OMI and A-Train observations of volcanic $SO_2$ loading and altitude for all significant volcanic eruptions (e.g., Carn, 2015; Carn et al., 2016) continue to play a crucial role in unraveling the major

natural sources of stratospheric AOD variability. OMI observations of reactive halogens (e.g., BrO, OClO) in volcanic eruptions clouds (e.g., Theys et al., 2014) also permit improved understanding of volcanic impacts on stratospheric ozone.

Since the first demonstration of OMI's ability to quantify volcanic $SO_2$ degassing (Carn et al., 2007, 2008), an important indicator of impending eruptions, OMI observations have been adopted by many volcano observatories worldwide as an essential tool for volcano surveillance. OMI data have subsequently been used to investigate or monitor volcanoes in most

volcanic regions of the globe, including East Africa (e.g., Sawyer et al., 2008; Ferguson et al., 2010; Goitom et al., 2015), Indonesia (Surono et al., 2012; Kushendratno et al., 2012), Papua New Guinea (McCormick et al., 2012), Vanuatu (Bani et al., 2009a, 2009b, 2012), Central America (Campion et al., 2012), the West Indies (Carn and Prata, 2010; Flower and Carn, 2015), Ecuador (Carn et al., 2008), Chile (Theys et al., 2014), Russia (Telling et al., 2015), Alaska (USA; Lopex et al., 2013) and Iceland (Sigmarsson et al., 2013; Schmidt et al., 2015). Satellite measurements of elevated $SO_2$ emissions (including

from OMI) were decisive in providing advance warning of a major eruption at Merapi (Indonesia) in 2010, permitting evacuation of the flanks of the volcano and saving many lives (Surono et al., 2012).

Recent improvements in $SO_2$ algorithm sensitivity (Li et al., 2013, 2016; Theys et al., 2015) have increased the sensitivity of OMI measurements to weak volcanic $SO_2$ degassing. Coupled with new $SO_2$ emission estimation techniques (Fioletov et al., 2011), these data have permitted the identification of ~100 volcanic $SO_2$ emission sources (roughly two-

thirds of the ~150 degassing volcanoes known worldwide), including some sources in remote regions with no prior measurements, which will comprise a new volcanic $SO_2$ emissions inventory (Fioletov et al., 2016; Carn et al., 2017). This inventory will improve constraints on volcanic emissions of other important gases, such as $CO_2$, which are difficult to measure directly.

A significant increase in demand for near real-time (NRT) satellite observations of volcanic clouds occurred following

the Eyjafjallajökull eruption in Iceland in April-May 2010, which disrupted aviation operations on a global scale. UV measurements have some unique advantages for volcanic ash detection, notably the ability to detect ash (and $SO_2$) when located above or mixed with clouds (e.g., Carn et al., 2009; Carn and Krotkov, 2016). Timeliness of observations and data delivery is critical for aviation safety, and this has be optimized by using satellite Direct Broadcast/Readout (DR) capabilities. The operational OMI VFD (Very Fast Delivery) service (Hassinen et al., 2008) for NRT detection of volcanic

$SO_2$ and ash emissions was implemented at FMI in 2011. The existing Satellite Measurements from Polar Orbit (SAMPO; http://sampo.fmi.fi/volcanic.html) service utilizes the DR capability of the Aura/OMI and SNPP/OMPS instruments and the ozone, cloud reflectivity, volcanic $SO_2$ and Aerosol Index (AI) products are available to users, including the Support to Aviation Control Service (SACS; http://sacs.aeronomie.be/nrt; Brenot et al., 2014) and Volcanic Ash Advisory Centres (VAACs), within 20 minutes after the satellite overpass over a ground station.

The two ground stations, one in Sodankylä (northern Finland, used for both OMI and OMPS) and the second in Fairbanks (Alaska, USA, used for OMPS), ensure spatial coverage over busy airspace in the north Atlantic and north Pacific with many active volcanoes. The ongoing NASA Applied Sciences Project will enhance the Decision Support System services and tools used at VAACs by combining real time satellite DR observations with volcanic cloud dispersion modelling to provide improved forecasts of the $SO_2$ and ash together with the observations.

The Sodankylä VFD system proved its usefulness during two recent Icelandic eruptions: Grimsvötn in 2011 (Kerminen et al., 2011) and Holouraun- Bárðarbunga in 2014—2015 (Ialongo et al., 2015) as shown in Fig. 7. These recent eruptions also demonstrated that, in addition to aviation hazard mitigation, the OMI DR data could be used to anticipate and monitor air quality impacts due to low-altitude volcanic $SO_2$ and ash clouds.





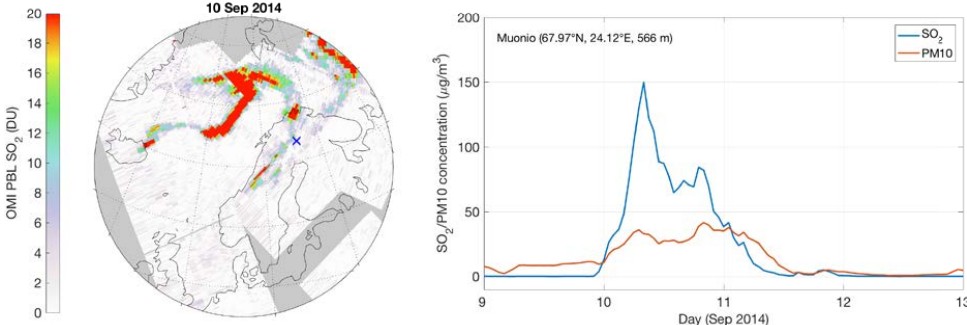

**Figure 7: Left: detection of SO₂ emissions from the Holuhraun (Iceland) eruption by the OMI Very Fast Delivery (VFD) system on 10th September 2014. Right: ground–based SO₂ measurements and breathable aerosols (PM10) over Muonio measurement station (Finland; location indicated by cross on VFD image) from 9 September up until and 13 September 2014.**

**5 Solar spectral irradiance monitoring**

OMI collects solar spectral irradiance (SSI) data primarily to provide long-term on-orbit calibration, in particular for characterization of throughput degradation and wavelength calibration. These goals have been met very well, as described in detail by Schenkeveld et al. (2017). Most OMI level 2 products use a constant solar irradiance reference spectrum to produce Earth reflectance data that are used in the actual retrieval processing. This 'fixed' irradiance spectrum comes from either an external high-resolution composite data set (e.g. Dobber et al., 2008a) or from the early epoch (2004-2005) OMI irradiance measurements. However, numerous observations show substantial solar variability (up to ~2% in strong spectral lines at the OMI spectral resolution, and larger for higher resolution measurements, see Fig. 8) for the spectral region below 300 nm on both solar rotational (~27-day) and solar cycle (~11-year) time scales (e.g., DeLand and Cebula, 2008). Similar variations are also clearly seen in the cores of selected Fraunhofer lines longward of 300 nm (Fig. 8). More importantly, results from the SIM (Spectral Irradiance Monitor) instrument on the SORCE (Solar Radiation and Climate Experiment) satellite (Harder et al., 2009) and their implications for climate response (Haigh et al., 2010) have led to ongoing debate regarding the magnitude of solar cycle variability across the entire OMI spectral region (e.g., DeLand and Cebula, 2012; Lean and DeLand, 2012; Ermolli et al., 2013; Morrill et al., 2014; Ball et al., 2016). Thus, developing an independent SSI data set from OMI has significant potential benefits for both solar physics and climate studies.





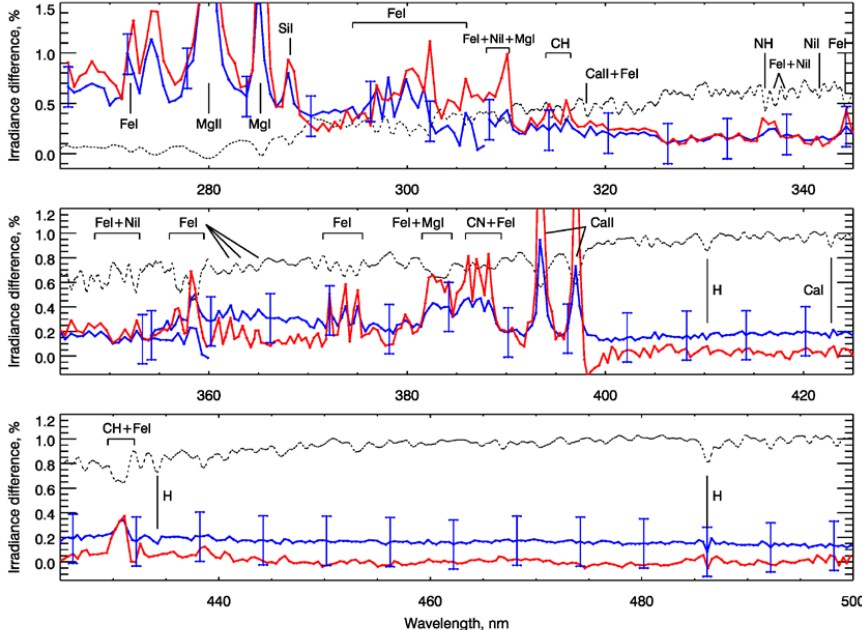

**Figure 8: F Solar Spectral Irradiance (SSI) variability in Cycle 24. Blue line shows the normalized long-term difference (2012-2014 vs. 2007-2009, i.e., the solar maximum vs. solar minimum) as observed by OMI, with representative ±0.2% errors. Red line follows the properly normalized GOME-2 data derived from the rotational (~27 days, the local maximum flux vs. the adjacent minimum) variability in 2012-2013. The GOME-2 data were adjusted to the OMI SSI by a multiplicative factor that matches the 27-day and Solar-Cycle amplitudes in the 325-335 nm range. Note that the ~3 times higher GOME-2 spectral resolution leads to substantially higher SSI amplitudes at prominent spectral lines and blends (e.g., CaII 393, 396 nm). For reference, the scaled solar spectrum is shown as dotted line.**

Creating a SSI data set from OMI measurements requires a comprehensive correction for instrument response degradation, which can have a complex spectral and temporal dependence (e.g. Floyd et al., 1998; DeLand and Cebula, 2008). A first step in this process is to demonstrate that OMI fully captures solar activity variations by creating a proxy index that is insensitive to most instrument degradation effects. This can be done using core-to-wing ratios for absorption

features such as Mg II $h$ and $k$ (280 nm), Ca II K (393.4 nm), and Ca II H (396.8 nm). DeLand and Marchenko (2013) describe the creation of Mg II index and Ca II index products from OMI irradiance data and show that these products agree very well with concurrent solar activity data sets from GOME (Global Ozone Monitoring Experiment), SORCE SOLSTICE (Solar Stellar Irradiance Comparison Experiment), and the National Solar Observatory. The OMI solar proxy data are available on-line at https://sbuv2.gsfc.nasa.gov/solar/omi/.

Since OMI does not carry an end-to-end onboard calibration system, some form of external information is necessary to develop a long-term degradation correction. Marchenko and DeLand (2014) observed that the extended solar minimum period between Cycle 23 and Cycle 24 (early 2007 to mid-2009), with no appreciable long-term (> 27-day) variation, began more than 2 years after the start of regular OMI solar measurements. Considering the extremely slow changes observed in OMI instrument throughput (Schenkeveld et al., 2017), we can reasonably expect that during such a period, the instrument

response changes should have settled into a linear time dependence. The next assumption, that all observed OMI long-term irradiance changes during the solar minimum period represent instrument response changes, then allows the derivation of spectrally dependent degradation rates. Correction functions based on these rates were used to produce calibrated OMI solar irradiance data for the period 2007-2013. The long-term solar cycle changes derived from the degradation-corrected OMI data agree very well with scaled short-term variations computed from contemporaneous GOME-2 data (Fig. 8). The spectral





dependence of the SSI variation observed by OMI for Cycle 24 is in excellent agreement with the dependence calculated using scaling factors based on Cycle 21-22 data (DeLand and Cebula, 1993).

Marchenko et al. (2016) extended this work and created accurate (~0.1-0.3% per 0.5-1 nm spectral bin), degradation-corrected daily OMI irradiance data. Since these data cover both the solar minimum and maximum, they may serve as

valuable, independent source for detailed comparisons with both SORCE measurements and calculations from the empirical Naval Research Laboratory Solar Spectral Irradiance, 2[nd] Version (NRLSSI2) (Coddington et al., 2016) and the semi-empirical Spectral And Total Irradiance Reconstructions (SATIRE-S) (Yeo et al., 2015) models. The behavior of SORCE measurements for Cycle 24 shows distinct differences from OMI observations and model predictions, particularly between 290-350 nm (Fig. 9). The OMI observations are very consistent with the NRLSSI2 results over the full spectral range of

OMI measurements.  The SATIRE-S predictions also show a good match to the OMI observations, although some differences are seen in regions occupied by strong spectral lines and line blends.  This agreement between OMI observations and widely used models provides an important constraint on solar variability in the near-UV and visible regions.

Continuation of these solar irradiance measurements by TROPOMI will be a valuable contribution in the coming years.

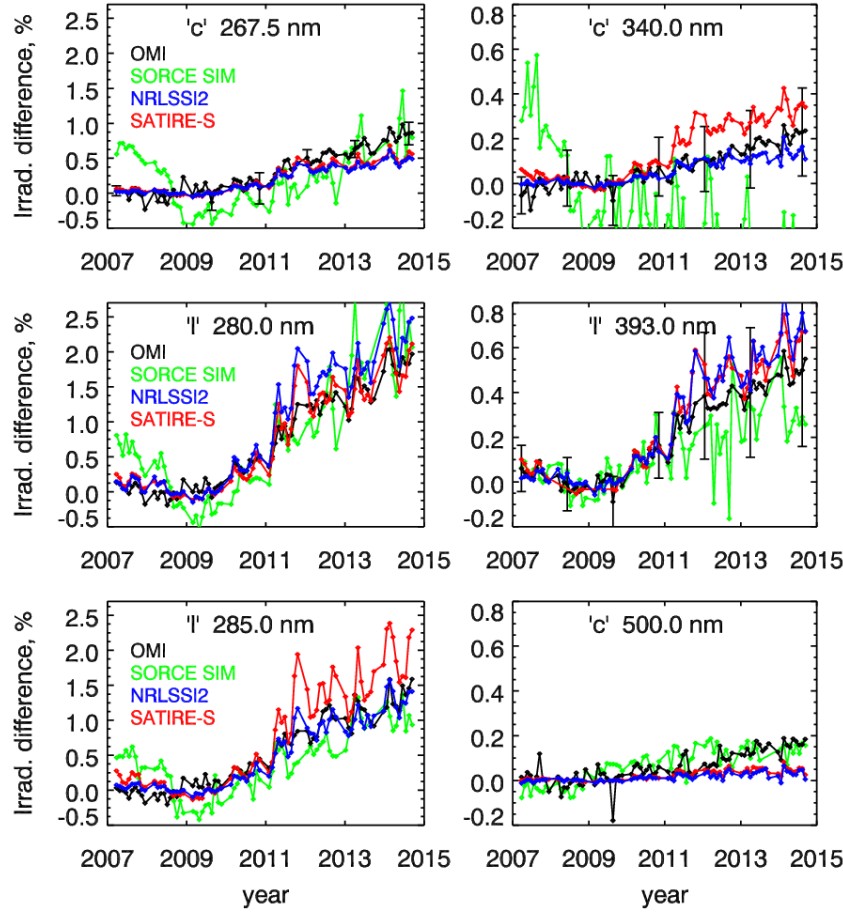

**Figure 9: The normalized, wavelength- and time-binned OMI irradiance changes relative to the 2009 solar minimum (with representative ±1-sigma error bars) compared to the consistently sampled model outputs and the SORCE SIM data. The strong spectral transitions are denoted with `l' and the relatively line-free regions with `c'.**




## 6 The Montreal Protocol, Total ozone, and UV-radiation

In 1987, the Montreal Protocol was established in an effort to protect the ozone layer. Ozone measurements by satellite are an important means to assess the Montreal Protocol's effectiveness to enable the ozone layer recovery from the effects of ozone depleting substances. NASA's contribution was a series of ozone monitoring instruments, beginning with the SBUV/
TOMS instrument on Nimbus 7 in 1978. The OMI instrument on Aura and a series of SBUV/2 instruments on NOAA satellites have continued this critical monitoring function. The Antarctic ozone hole has proven to be a very sensitive indicator of the state of the ozone layer. The high spatial resolution measurements, first by TOMS and continued by OMI, have been particularly important in mapping the development of the Antarctic ozone hole each year. These current and recent results are a prominent aspect of the quadrennial ozone depletion assessment that is written for the Parties to the
Montreal Protocol. In Fig. 10, the 2016 ozone hole as measured by OMI can be seen, based on the monthly average from October.

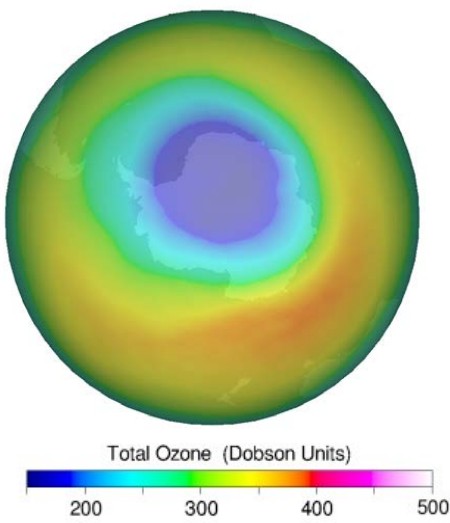

**Figure 10: October 2016 monthly average OMI total ozone column over Antarctica.**

### 6.1 The OMI Long Term Ozone Data Record

The data record of total column ozone from OMI has proven to be very stable over the ten plus years of operation. This stability is shown three ways (McPeters et al., 2015), by direct monitoring of instrument performance, by comparing OMI
ozone with that from ground based measurements, and by comparison with ozone from other satellite systems.

OMI instrument stability is monitored by tracking instrument parameters such as onboard measured solar flux (see Dobber et al., 2008b and Schenkeveld et al., 2017). Stability is also monitored by tracking changes in geophysical parameters like average ice reflectivity in Greenland and Antarctica. All these parameters show that OMI has been far more stable than any of the previous TOMS instruments. Two distinct algorithms have been used to compute total column ozone
from OMI, a TOMS-type algorithm and a Differential Optical Absorption Spectroscopy (DOAS) algorithm (Veefkind et al., 2006). A variation of the version 8 TOMS algorithm (Bhartia, 2007) used to process data from the series of TOMS instruments has been used for the OMI-TOMS retrieval. Designated the v8.5 algorithm, the most significant enhancement is that the longer wavelengths measured by OMI are used to infer cloud height on a scene-by-scene basis. OMI-TOMS ozone





results are shown here. A comparison of OMI-TOMS and OMI-DOAS total ozone products can be found in Kroon et al., 2008.

Comparisons with ground based observations show that OMI has been very stable (see Fig. 11). Such comparisons have been shown capable of detecting instrument changes of a few tenths of a percent (Labow et al., 2013). The linear fit in Fig. 5    11 shows that OMI has almost no drift in ozone relative to the ground observations (0.05% per decade). The offset of about −1.5% is mostly caused by the use of the older Bass and Paur (1984) ozone cross sections in the OMI retrievals rather than the newer Brion/Daumont/Malicet ozone cross sections (Brion et al., 1993).

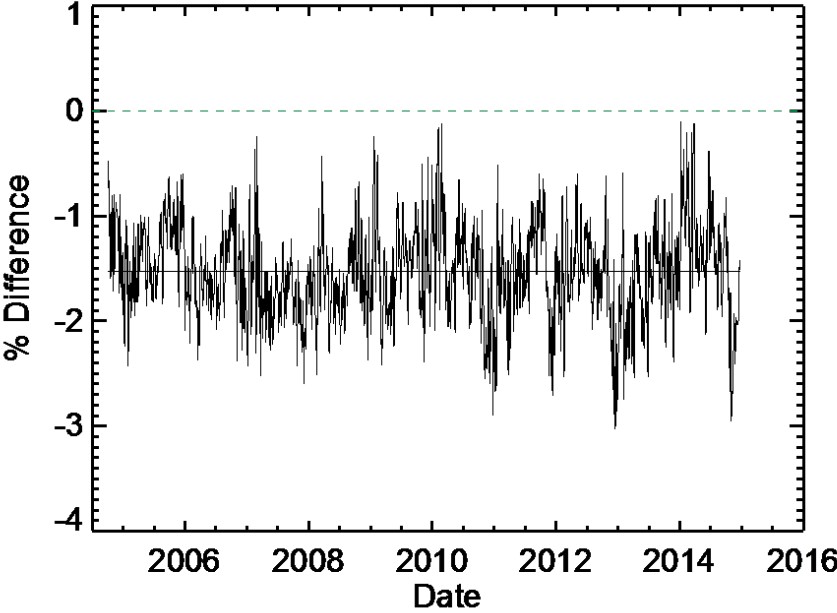

**Figure 11: Weekly mean percent difference of OMI ozone minus ground-based averages from an ensemble of 76** 10   **Northern Hemisphere Dobson–Brewer stations with linear fit (from McPeters et al., 2013).**

Figure 12 shows comparisons of OMI ozone with data from a series of SBUV/2 instruments flying on NOAA spacecraft. The radiances from SBUV instruments on NOAA 16, 17, 18, and 19 were carefully analyzed and adjusted to create a consistent ozone data series. Here global average ozone from 60° S to 60° N is plotted as well as percent difference for each 15   instrument. The first thing to note is the high degree of consistency of the four NOAA instruments. There is a small trend of OMI relative to SBUV of about +0.4% per decade and an average bias of -0.9 %. While this trend might be considered statistically significant, at the half percent per decade level it is not possible to say whether one trend is more accurate than the other.





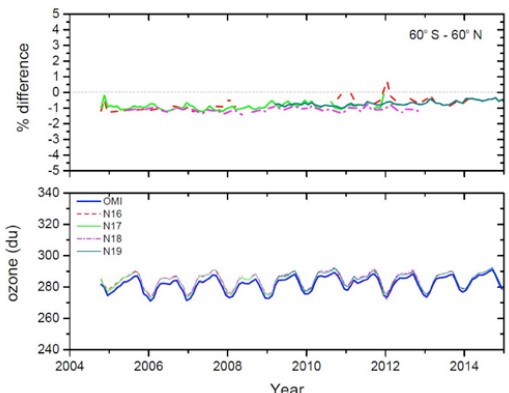

**Figure 12: Total column ozone from OMI and four NOAA SBUV instrument (bottom) and differences in % of OMI minus each SBUV instrument (top) (adapted from McPeters et al., 2013).**

Similar comparisons with ozone from instruments on European satellites can be used to see if the behavior of OMI ozone displays similar patterns. The GTO (GOME Type Ozone) merged ozone data record is based on data from sensors on GOME/ERS-2 (1995–2011), SCIAMACHY/ENVISAT (2002–2012), and GOME- 2/METOP-A (2007–present). The GTO analysis combines these measurements into a continuous and homogeneous monthly mean time series (Coldewey-Egbers et al., 2015). In Fig. 13 OMI ozone averaged from 60° S to 60° N is compared with the v8.6 MOD (Merged Ozone Data) time

series based on a best effort merger of the NASA SBUV/2 data shown in Fig. 12 (Frith et al., 2014) and with this GTO time series. The OMI bias relative to GTO is a bit larger, -1.7% vs. -1.0% for MOD over the same time period, again mostly due to cross section differences. OMI has a small positive trend relative to MOD over the 2004–2011 time period, and a small negative trend relative to GTO of -0.85% per decade. Given the difficulty of maintaining long term calibration of multi-instrument data sets, differences of 1% or so per decade are probably the best one can do, and these differences should be

considered within the range of uncertainty.

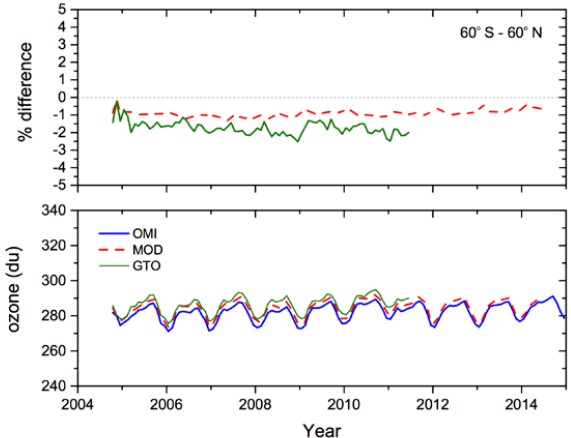

**Figure 13: Total column ozone from OMI, the MOD (merged ozone data) based on SBUV/2 instruments, and the**
**GTO (GOME type Total Ozone) merged ozone based on GOME instruments and SCIAMACHY (bottom) and differences in % of OMI minus each (top) (from McPeters et al., 2015).**





OMI total ozone column measurements have also been used as integral part of the Multi Sensor Reanalysis (MSR) datasets versions 1 (MSR-1; van der A et al., 2010) and version 2 (MSR-2; van der A et al., 2015) dataset. The MSR-2 data set is a 43-year total ozone column assimilation data set for 1970-2012 based on a multitude of satellite instruments

measuring total ozone columns and provides global daily fields of total ozone columns. In Fig. 14, a time series is given with ozone hole images and the corresponding September mean total ozone column over Antarctic based on the MSR-2 assimilated total ozone column data [van der A et al., 2015] for 1979-2012 in blue/red, and OMI assimilated total ozone columns for 2013-2016 indicated in grey. Images and daily total ozone columns data are obtained from the TEMIS website (http://www.temis.nl). The red colors denote the six years that are known to have been disturbed by naturally occurring

planetary wave activity, which lead to reduced seasonal Antarctic stratospheric ozone destruction (de Laat et al., 2017, and references therein; JGR - under review). The MSR datasets have been used to monitor atmospheric processes affecting long term local total ozone column variability (Knibbe et al., 2014) as well as long term changes in Antarctic stratospheric ozone (Ozone Hole) and recovery (Knibbe et al., 2014; de Laat et al., 2015, 2017). Detection of recovery of Antarctic stratospheric ozone has turned out to be complicated due to ambiguities in Antarctic Ozone Hole metrics and analysis methods (Knibbe et

al., 2014; de Laat et al., 2015). However, de Laat et al. (2017), using more robust Antarctic Ozone Hole metrics like the Ozone Mass Deficit, show that post year-2000 Antarctic stratospheric ozone recovery appears to be well under way. After reaching maximum ozone destruction around the year 2000, current levels of seasonal Antarctic stratospheric ozone destruction appear to have returned to early 1990s levels. In addition, the MSR data are also used for the annual WMO Antarctic ozone bulletins that provide regular seasonal analyses of the status of the Antarctic Ozone Hole of that particular

year (e.g., Braathen et al., 2015). Note that the MSR-1 dataset has also be used to provide a global daily surface clear-sky UV index dataset.

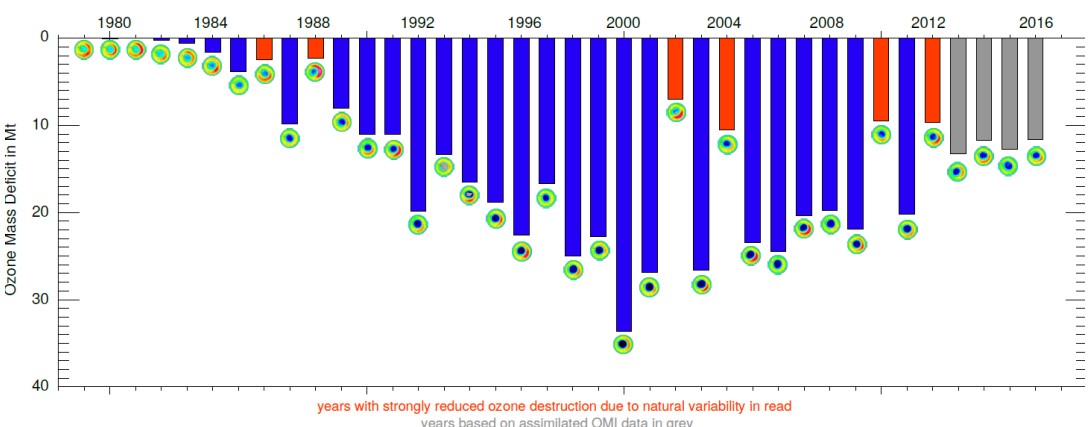

**Figure 14: Annual September average daily Ozone Mass Deficit over Antarctic relative to the 220 DU total ozone column level.**

**6.2 Global Surface UV radiation**

Surface UV estimates based on OMI satellite data continue the long-term TOMS UV record. The OMI UV algorithm

(Tanskanen et al., 2006) was further developed from the TOMS algorithm (Eck et al., 1995; Krotkov et al., 1998, 2001). It consists of a calculation for the clear sky case with corrections for clouds (or non-absorbing aerosols). Several validation




studies of both TOMS and OMI-UV data have shown a positive bias in many locations affected significantly by absorbing aerosols (e.g., Arola et al., 2005; Tanskanen et al., 2007, Zempila et al., 2016). Correction for the absorbing aerosols was suggested by Arola et al. (2009), which exploits monthly aerosol climatology of Kinne et al. (2013). This correction is currently also implemented in the OMI UV product. Figure 15 gives an example of the OMI UV product, showing the long-

term seasonal mean of fall season (September-November) UV index, calculated from the 10-year OMI record (2005-2015). OMI UV data records have also provided valuable information for UV chapters of the WMO Scientific Assessment of Ozone Depletion both in 2006 and 2010 (Bais et al., 2007, Douglass et al., 2011).

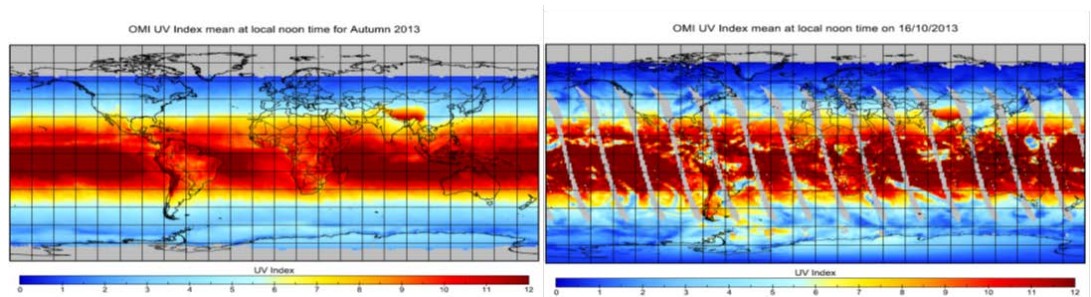

**Figure 15: On left: Three-month mean UV Index from OMUVB in the boreal fall season 2013 (September-November). On right: Global map of daily UV Index on 16th of October, 2013 showing exceptionally high UV Index values in Patagonia due to the stretched ozone hole.**

**7 Tropospheric ozone from OMI: Overview of different methods**

Tropospheric ozone is an important pollutant at ground level, plays a critical role in oxidation and atmospheric chemistry, and is a greenhouse gas in the upper troposphere. As the retrieval of tropospheric ozone is a challenging (strongly ill posed) task, several approaches have been developed to overcome the challenges. OMI has fostered a large number of tropospheric ozone data products, both as ozone column amounts and ozone profiles. These products, as shown in Fig. 16, have been developed using either OMI measurements alone or in conjunction with other satellite measurements to improve sensitivity

to near surface ozone (e.g., Bowman, 2013; Cuesta, et al., 2013; Hache et al., 2014) as summarized below. They have been used in tropospheric research (e.g., Sauvage et al., 2007; Ziemke et al., 2010; Cooper et al., 2014), for example to show evidence of decadal increases/trends in global tropospheric ozone, El Nino events during Aura (e.g., Chandra et al., 2009; BAMS State of the Climate report for year 2015), the 1-2 month Madden-Julian Oscillation (Ziemke et al., 2015, and references therein), and urban pollution (Kar et al., 2010).



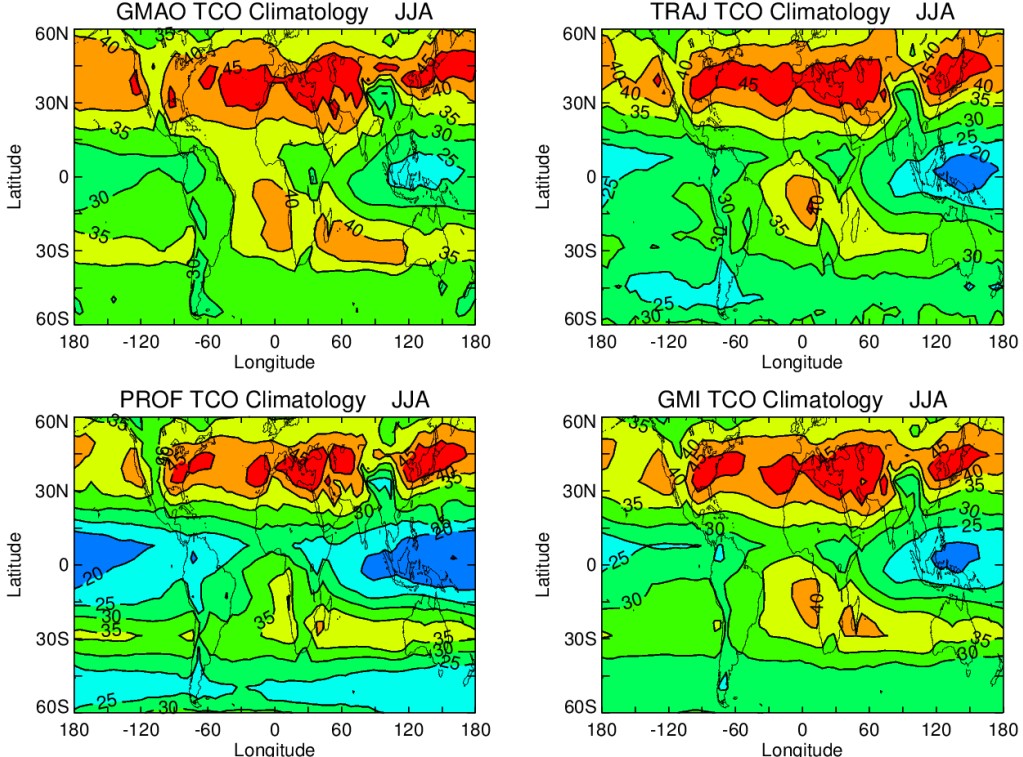

**Figure 16: June-July-August (JJA) seasonal climatology of tropospheric column ozone (in Dobson units) for three OMI/MLS products and the Global Modeling Initiative (GMI) chemical transport model. ASSIM denotes assimilated OMI/MLS, TRAJ is trajectory-mapped OMI/MLS, and PROF is OMI-only profile retrieval). This figure is from Ziemke et al. (2014) which includes references and detailed descriptions for these three products and the GMI model.**

### 7.1 Cloud Slicing

The convective-cloud differential (CCD) method (Ziemke et al., 1998) uses the differences between OMI total column ozone and OMI above-cloud column ozone under conditions of high reflectivity (i.e., deep convective clouds) to estimate a tropospheric column ozone residual. The CCD algorithm is simple to apply but not very effective for measuring tropospheric ozone outside the tropics. Measuring tropospheric ozone outside tropical latitudes is possible to accomplish by using either an OMI-only profile algorithm, or the neural network approach (Sellitto et al., 2011), or by combining OMI with another satellite measurements. Some of these are discussed in more detail below.

### 7.2 Profile Retrieval Algorithms

Strong spectral variation of both ozone absorption (decrease by ~4 orders of magnitude from the Hartley to Huggins bands) and Rayleigh scattering ($\sim\lambda^{-4}$) lead to wavelength-dependent photon penetration, and therefore provide vertical discrimination of ozone in the atmosphere (Bhartia et al., 1996). Temperature-dependent ozone absorption in the Huggins bands adds additional tropospheric ozone information (Chance et al., 1997). Based on these principles, two ozone profile algorithms were implemented: the operational algorithm (OMO3PR) at KNMI (Kroon et al., 2011; Mielonen et al., 2015) and the research algorithm (PROFOZ) at the US Smithsonian Astrophysical Observatory (SAO) (Liu et al. 2010a, b; Kim et al., 2013). Both retrieve ozone profiles from the spectral region 270-330 nm using the optimal estimation method, but they differ significantly in implementation details including radiometric calibration, radiative transfer model simulation, a priori





constraint, retrieval grids, and retrieval parameters. Typically, the retrievals have 5-7 degrees of freedom (DOF) for signals for ozone with up to ~1.5 DOF in the troposphere. It has been shown that tropospheric ozone column can be directly and accurately retrieved in the few Dobson Units range from OMI data alone on the spatial pixel-to-pixel basis, but successful retrievals of tropospheric ozone and further capture of tropospheric ozone trends require accurate forward model simulation,

well-characterized prior information, and consistently accurate radiometric calibration over the entire record (Liu et al., 2010a, 2010b; Mielonen et al., 2015).

Validation of the OMO3PR product by Kroon et al. (2011) showed that the operational retrieval agrees well with global correlative datasets to within 20% but with some biases. These biases can be slightly reduced using a different a priori constraint and surface albedo assumptions; most of the biases are likely caused by systematic biases in radiative-transfer

modeling and radiometric calibrations (Mielonen et al., 2015). Tang et al. (2012) indicated that this product has some skill in identifying stratosphere-troposphere folds. The methods employed by the TROPOMI ozone profile algorithm will be used to update the current OMI ozone profile algorithm (de Haan, 2015).

For PROFOZ, early versions were partially evaluated against ozonesonde, aircraft, MLS, surface measurements, model simulations, and tropospheric ozone derivations from other methods (Pittman et al., 2009; Liu et al., 2010b; Zhang et al.,

2010; Walker et al., 2010; Tarasick et al., 2010; Sellitto et al., 2011; Wang et al., 2011, 2013; Bak et al., 2013a,b; Liu et al., 2013; Flynn et al., 2014; Foret et al., 2014; Ziemke et al., 2014). The analyses generally showed good agreement with other correlative data but revealed limited sensitivity to ozone in the lower troposphere and near the surface. In addition to these evaluations, the PROFOZ product has been used to study dynamical and chemical features associated with stratospheric-tropospheric exchange, to evaluate the transport of anthropogenic pollution (Pittman et al., 2009; Liu et al., 2010a; Walker et

al., 2010; Su et al., 2011; Liu et al., 2013), to constrain tropospheric ozone sources (Zhang et al., 2010; Kim et al., 2013), to initialize boundary conditions for air quality modeling (Pour-Biazare et al., 2011), and to study ozone enhancement in the lower troposphere over central and eastern China (Hayashida et al., 2015, 2016).

### 7.3 Multi-instrument retrievals

The Aura MLS measures ozone profiles along orbital track from the top of the atmosphere down to the tropopause/upper

troposphere. Several schemes have been used to derive tropospheric ozone by combining OMI and MLS. Jing et al. (2006) and Ziemke et al. (2006) subtracted MLS stratospheric column ozone from OMI total column ozone to derive tropospheric column ozone residual. While Jing et al. (2006) applied a criterion for near-coincidence between OMI and MLS along orbital-track, Ziemke et al. (2006) used a 2D interpolation technique to fill in missing MLS measurements between orbital tracks and improve horizontal coverage. Schoeberl et al. (2007) further used a wind trajectory mapping technique of MLS

ozone profiles and Yang et al. (2007) used potential vorticity mapping to obtain better signal-to-noise and horizontal coverage for the OMI/MLS tropospheric column ozone. Wargan et al. (2015) discusses an OMI/MLS tropospheric ozone profile product derived using data assimilation; it is noted that current MERRA-2 analyses include tropospheric ozone profiles determined similarly via data assimilation of Aura MLS and OMI ozone. A comparison of several OMI/MLS tropospheric column ozone products (data assimilation, trajectory mapping, and profile retrieval methods) is described by

Ziemke et al. (2014). They concluded that the assimilation was overall the best science product when considering temporal and spatial coverage and ability to provide an entire ozone profile for both troposphere and stratosphere.

Theoretical studies point towards the potential of combining UV Hartley-Huggins (270-330 nm) and TIR $O_3$ (9.6 μm) bands for retrieving ozone profiles (Landgraf and Hasekamp, 2007; Worden et al., 2007). The physical basis for the improved resolution is that the reflected sunlight radiances are sensitive to the tropospheric column whereas the TIR

sounders are primarily sensitive to the free-troposphere. The "subtraction" of the free tropospheric column from the total column results in an estimate of near-surface concentrations. The theory has been demonstrated by a suite of retrieval algorithms: GOME-2/IASI (Cuesta et al., 2013) and OMI/TES (Fu et al., 2013) for ozone profile retrievals. The MUlti-



SpEctra, MUlti-SpEcies, Multi-SEnsorS (MUSES) tropospheric ozone retrieval algorithm is implemented to extend the joint TES/OMI retrievals to the AIRS/OMI combination (Fu et al., 2016).

## 8 Research Data Products

Several new products have been developed after launch that were not part of the initial suite of standard products described in Levelt et al. (2006b). Here we describe some of these research and new standard products. Most of these are available through the Aura Validation Data Center (AVDC), https://avdc.gsfc.nasa.gov.

### 8.1 Aerosol Above Cloud

Contrary to the known cooling effects of these aerosols in cloud-free scenarios over dark surfaces, the overlapping situation of absorbing aerosols over cloud can potentially exert a significant level of atmospheric absorption and produces a positive radiative forcing (warming) at top-of-atmosphere. The magnitude of direct radiative effects of aerosols above cloud directly depends on the aerosol loading, microphysical, and optical properties of the aerosol layer and the underlying cloud deck and geometric cloud fraction. The optical depth of carbonaceous and desert dust aerosol layers located above clouds (ACAOD) has been retrieved with OMI (Torres et al., 2012) leading to a global daily product spanning the OMI record (OMACA, OMI ACAOD). OMACA can be used to improve our understanding of aerosol-cloud interaction. OMACA provides both the above-cloud aerosol optical depth as well as the optical depth of the underlying clouds layer using OMI measurements at 354 nm and 388 nm (Jethva et al., 2014b, 2016). Evaluation of the product using high-quality measurements from the first phase of the NASA Earth Ventures-Suborbital (EV-S) ObseRvations of Aerosols above CLouds and their intEractionS (ORACLES) field campaign is in progress.

### 8.2 Water Vapor Column

Water vapor has a set of absorption bands in the visible region of the spectra. Despite being much weaker than other bands at longer wavelengths, they can be used to retrieve water vapor from OMI. A new column water vapor product (OMH2O) has been developed, evaluated, and implemented (Wang et al., 2014, 2016). The product uses a spectral fitting window of 430-480 nm and compares well with other data sets including MODIS, GPS, and AERONET.

### 8.3 Glyoxal Column

Glyoxal (CHO-CHO) is a short-lived product of non-methane volatile organic compound (NMVOC) atmospheric oxidation, a process that is important for both air quality and climate. The combination of formaldehyde and glyoxal measurements can provide useful information on NMVOC speciation (Di Gangi, 2012, Chan Miller et al., 2016) and can constrain NMVOC emissions (Stavrakou et al., 2009). Glyoxal has been retrieved from OMI (Chan Miller et al., 2014, 2016) using wavelengths 435-461 nm. The retrieval of glyoxal is challenging due to its very weak absorption (optical depths on the order of $10^{-4}$-$10^{-3}$). The OMI glyoxal research product is optimized to minimize interferences from stronger absorbers. The retrieval consists of three steps, (1) slant column density (SCD) fitting in the visible spectral region (435 nm – 461 nm), (2) air mass factor calculations to convert SCDs in to vertical column densities, and (3) de-striping using a reference sector over the Sahara (Chan Miller et al., 2014). Nearly ten years of glyoxyl data (up to 2014) are available. Given the challenging nature of glyoxal retrievals, detector degradation affects the quality of the retrievals after 2013 (Chan Miller, 2016).



### 8.4 NO$_2$ Cloud Slicing

The use of cloud pressure information from OMI has led to so-called cloud slicing approaches to retrieve profile information about trace gases. While this approach has most commonly been applied to ozone (see Sect. 7), it has also been applied to NO$_2$ with OMI to derive information about its concentration in the free troposphere (Choi et al., 2014; Belmonte

Rivas et al., 2015). In addition, other approaches have been applied to isolate NO$_2$ generated from lightning (Bucsela et al., 2010; Pickering et al., 2016).

### 8.5 Specialized Data Sets

As the first generation of OMI trace gas products typically used static databases for profile information, specialized products emerged to meet various needs of the user community. Several examples apply to NO$_2$ and SO$_2$ and in some cases

these were enabled by information provided in the standard products (e.g., Lamsal et al., 2008, 2015; Yang et al., 2010; Lee et al., 2011; Russell et al., 2011, 2012; McLinden et al., 2014; Theys et al., 2014; De Smedt et al., 2015). At the same time, this research has driven improvements in the standard products.

### 8.6 Polar Mesospheric Clouds (PMCs)

Another valuable but unplanned data product is the detection and characterization of polar mesospheric clouds (PMCs).

These clouds (also called noctilucent clouds) are observed at 80-85 km altitude and high latitudes (> 50°) during summer in each hemisphere and are potentially another indicator of long-term climate change (Thomas, 1996). Backscatter ultraviolet (BUV) instruments such as OMI detect PMCs as an enhanced signal at short wavelengths (DeLand et al., 2010). The broad cross-track coverage of OMI makes it possible to directly characterize local time variations in PMC occurrence frequency and intensity (DeLand et al., 2011). PMCs can also affect derived values of profile ozone in the upper stratosphere, so that a

correction is required to obtain better results (Bak et al., 2016).

### 9 Multi-platform Product and Analyses using Several Instruments across Platform(s)

The development of the so-called "A-train", a constellation of satellites in a common afternoon orbit all flying within about 15 minutes of each other, has provided unique opportunities to combine data from different instruments into new products, to incorporate additional information to enhance existing OMI products, and to cross validate other products with

OMI. Among the satellites used in conjunction with the OMI, the A-train includes the NASA Aqua satellite, that hosts the MODerate-resolution Imaging Spectroradiometer (MODIS), AIRS, and the Clouds and the Earth's Radiant Energy System (CERES), the Cloud-Aerosol Lidar and Infrared Pathfinder Satellite Observation (CALIPSO), a joint US (NASA) and French (CNES) satellite mission that includes the Cloud-Aerosol Lidar with Orthogonal Polarization (CALIOP) instrument, the millimeter-wavelength cloud radar aboard the NASA CloudSat, and the CNES Polarization & Anisotropy of

Reflectances for Atmospheric Sciences coupled with Observations from a Lidar (PARASOL) that carried the Polarization and Directionality of the Earth's Reflectances (POLDER) instrument that was operational from 2004-2013. Data from other satellites not in a common orbit provide additional opportunities to enhance and eventually extend OMI data.

In addition to the examples provided below, there are numerous works that employ cross platform comparisons for evaluation of OMI and other satellite data sets and algorithms., For example, Veefkind et al. (2011) used spatial and

temporal correlations between concurrent satellite observations of aerosol optical thickness (AOT) from the Moderate Resolution Imaging Spectroradiometer (MODIS) and OMI tropospheric columns of NO$_2$, SO$_2$, and HCHO to infer information on the global composition of aerosol particles. Other studies that use cross-platform data sets together for synergetic analyses for volcanic eruption and studies of air pollution e.g., Carn et al., 2007; Witte et al., 2011; Hsu et al.,



2012; Wang et al., 2013). Studies specifically for aerosol include Carboni et al., (2012), Chen et al., (2012) and Jethva et al., (2014) and Zhu et al., (2016) for HCHO. Examples of combining ozone information from different platforms are given in Sect. 7.3.

### 9.1 OMI Field-of-View for Collocation

To aid in the interpretation of OMI data and use them in conjunction with other instruments, it is important to have a precise estimate of its field-of-view (FOV). To this end, in depth comparisons of collocated OMI and MODIS radiances have been conducted (de Graaf et al., 2016; Sihler et al., 2016). Results show that the OMPIXCOR product 75FoV corner coordinates are accurate as the full width at half maximum (FWHM) of a super-Gaussian FOV model when this function is assumed. These studies are anticipated to help expand the work of de Graaf et al. (2012) by allowing for the use of collocated OMI and MODIS data to compute the aerosol direct effect over clouds among other applications.

### 9.2 A-train Collocated Products

The OMI team has developed collocated products that aid in algorithm development and validation. These include a new standard product that contains both OMI cloud products as well as many Aqua MODIS statistical cloud parameters collocated to OMI footprints known as OMMYDCLD. Over the past two years, working closely with aerosol algorithm developers, the team also produced a new level 2 orbital track product (OMMYDAGEO) that collocates OMI geo-coordinates (row and scan number) onto the MODIS granule at 3 and 10 km scales. This product assists users with the computationally burdensome task of collocating data from these two instruments, providing a direct link between the MODIS and OMI aerosol data products at two different spatial resolutions.

### 9.3 Aerosol Products

Aerosol products have benefitted from the A-train in several ways. For example, in the OMAERUV UV aerosol product, data from the CALIOP have been used to constrain the aerosol layer heights, and carbon monoxide (CO) data from AIRS have been used to help distinguish different types of absorbing aerosol, i.e., smoke from dust (Torres et al., 2013). MODIS data (OMMYDCLD) have been used to evaluate effect of subpixel cloud contamination (Gassó and Torres, 2016). Aerosol optical thicknesses from MODIS have also been combined with OMI measurements to estimate aerosol layer height (Satheesh et al., 2009; Chimot et al., 2016). Another important science application of the OMI and A-train aerosol products is the first global estimate of shortwave direct radiative effect of aerosols at the top of the atmosphere (TOA-DREA) over land and ocean (Lacagnina et al., 2016). This work was carried out using data based on global satellite observations of SSA, phase function, and AOD from PARASOL, in synergy with OMI SSA retrievals. Aerosol information from these two sensors is combined with land-surface BRDF and cloud properties from MODIS to produce monthly mean TOA-DREA global monthly averages in 2006. The estimated global mean TOA-DREA is $-4.6 \pm 1.5$ W/m$^2$ for cloud-free and $-2.1 \pm 0.7$ W/m$^2$ for all-sky conditions. All-sky TOA-DREA is less negative than its cloud-free counterpart, because of enhanced planetary albedo by clouds and cloud masking effects on aerosol radiation interactions. These are the first DREA estimates constrained by satellite-based aerosol absorption observations.

The instantaneous TOA-DREA over clouds can be obtained by combining level 1 radiance measurements in the shortwave from OMI with radiance measurements from MODIS on the A-Train. The instantaneous TOA-DREA over clouds can be estimated very accurately using hyper spectral radiances of aerosol and clouds scenes (de Graaf et al., 2014), which can be achieved by combining OMI and MODIS radiances. The instantaneous TOA-DREA over clouds can reach values up to $130 \pm 8$ W/m2, which results in strong warming of the atmosphere at the location of the aerosol layer.



**9.4 Clouds and Radiation**

The two OMI cloud algorithms are based on oxygen dimer absorption at 477 nm (Acarreta et al., 2004; Veefkind et al., 2016) and rotational-Raman scattering at 350 nm (Joiner and Vasilkov, 2006), both related to photon path lengths in the atmosphere (Stammes et al., 2008). The A-train has provided unique opportunities to help interpret and evaluate these

measurements. Radiative transfer calculations using collocated cloud extinction profiles from MODIS and CloudSat have been used to evaluate the OMI retrievals (Vasilkov et al., 2008). In addition, a third photon path length type measurement (from PARASOL measurements of oxygen absorption in the A-band) provided additional measurements for evaluation (Sneep et al., 2008).

These works demonstrated that the cloud pressures derived from OMI and similar path length type measurements do not

measure the physical cloud top but rather an average pressure reached by solar photons inside a cloud. This pressure has been referred to as the optical centroid cloud pressure (OCCP) where the centroid relates to the vertical distribution of cloud reflectance. This led to the development of fast simulators that can be used to estimate OCCP based on vertical extinction profiles from either models or measurements such as those provided by CloudSat/MODIS (Joiner et al., 2012).

The OMI OCCPs were shown to be distinct from estimates of the physical cloud top provided by infrared, radar, or lidar

(e.g., Joiner et al., 2006; Ziemke et al., 2009; Avery et al., 2010). This then led to the development of an approach to detect multi-layer clouds using OMI OCCP in combination with cloud top pressures from Aqua MODIS (Joiner et al., 2010). Vasilkov et al. (2010) showed that optically thick clouds over snow and ice can be detected using the difference between retrieved OMI OCCP and the surface pressure. Finally, it has been noted that the effective cloud fraction (ECF), a standard parameter in the OMI cloud products, is nearly linearly related to top-of-the-atmosphere short-wave radiative flux (TOA-

SWF) (Gupta et al., 2016). They used nearly coincident estimates of TOA-SWF from the Aqua CERES along with OMI and other ancillary parameters to train an artificial neural network (ANN) to estimate TOA-SWF. This relationship is also exploited by the surface solar irradiance product from OMI (http://www.temis.nl/ssi), which is validated against the globally distributed Baseline Surface Radiation Network (BSRN) measurements (Wang et al., 2014).

**9.5 Trace gases**

Besides the work on volcano monitoring (Sect. 4) and estimated emissions using data from multiple platforms (Sect. 3.4), deriving trace gas concentrations from polar-orbiting satellite platforms that have different equator crossing times can provide information about diurnal variability. It is important that these types of cross platform analyses use a common algorithm. This has been accomplished for $NO_2$ using OMI and SCIAMACHY (Boersma et al., 2008) and formaldehyde (HCHO) using OMI and GOME-2 (De Smedt et al., 2015).

Information from satellites that observe in different geometries (limb and nadir views) may be combined to provide improved profile estimates. In addition to the work using multiple sensors with OMI to provide information on tropospheric ozone (see Sect. 7.3), Adams et al. (2016) have applied a variant of the limb–nadir matching technique for deriving tropospheric $NO_2$ columns from a combination of non-coincident profiles from the Optical Spectrograph and InfraRed Imaging System (OSIRIS) and slant column densities from OMI along with a photochemical box model to account for

diurnal variations of stratospheric $NO_2$ and the temporal mismatch in observations. This work shows the potential to combine information from polar and geostationary platforms.

**9.6 Geometry-dependent Lambertian Equivalent Reflectivity**

For most OMI algorithms, it is important to have accurate estimates of surface reflectance. Surface reflectance is complex because it varies with the sun-satellite viewing geometry as well as with time and space. Vasilkov et al. (2016)

constructed a global time-varying geometry-dependent Lambert-equivalent reflectivity (GLER) product (i.e., for each OMI



pixel) based on MODIS data and ocean models. This allows for integration into existing algorithms based on LER models without any major modifications to the algorithms. The GLER was tested within OMI $NO_2$ and cloud retrievals and found to have significant impact (Vasilkov et al., 2016). The GLER approach can also be applied to other UV-Vis instruments.

## 10 Aircraft and Other Field Campaigns

OMI data have been used to support flight planning for many international field campaigns and conversely data from these campaigns has helped to validate OMI retrievals. Here, we discuss two field campaigns relevant to OMI $NO_2$ and $O_3$ retrievals: Deriving Information on Surface Conditions from Column and Vertically Resolved Observations Relevant to Air Quality (DISCOVER-AQ) and the Cabauw Intercomparison of Nitrogen Dioxide Measuring Instruments (CINDI and CINDI-2) experiments described below.

**10.1 DISCOVER-AQ**

The DISCOVER-AQ project was a four-year NASA Earth Venture Sub-orbital (EV-S) mission to improve the use of satellites to monitor air quality for public health and environmental benefit. Through targeted airborne and ground-based observations, DISCOVER-AQ aimed at improving the interpretation of satellite observations to diagnose near-surface conditions relating to air quality. The first of four deployments took place over the Washington, DC and Baltimore, MD
metropolitan area throughout the month of July 2011. A wintertime campaign was conducted in the San Joaquin Valley of California aimed primarily at particulate matter. Warm season campaigns followed in Houston, Texas in September 2013 and in the Denver/Front Range region of Colorado in July/August 2014. Two aircraft were used, the NASA P-3B for in-situ sampling typically from 0.3 to 3 km altitude in spiral ascents and descents and the NASA UC-12 flying at ~8 km with remote sensing instruments for trace gases and aerosols. In addition, extensive ground observations were used to measure air
pollution at the surface using in-situ observations, and aloft, using balloons and remote sensing instruments. As part of these campaigns, a network of Pandora spectrometers provided continuous (every 20 s), high-resolution measurements of total column $NO_2$ and $O_3$ amounts at a minimum of twelve urban and rural locations characterized by different levels of pollution.

The OMI standard product tropospheric $NO_2$ retrieval Version 2.1 was compared with the vertical integration of the Baltimore/Washington DISCOVER-AQ P-3B aircraft data and the ground-based Pandora retrievals (Lamsal et al., 2014).
Data were used from within 1 hour of the OMI overpass. The aircraft data were extrapolated to the surface using a model-derived vertical gradient for each site. OMI agreed with the aircraft data to within ±20% in 60% of the cases at four of six sites, while there was a greater difference at two sites near coastal areas that have complex vertical and horizontal distributions of $NO_2$. On average the OMI columns were less than those from the aircraft by 6 to 22% except at the two coastal sites, where the aircraft column was often a factor of two greater. OMI total column $NO_2$ was less than Pandora by
<6% at three sites and greater than Pandora by 9-13% at two sites.

Pandora total columns of $O_3$ and $NO_2$ were compared with those from OMI at all 12 sites in the Baltimore/Washington region by Tzortziou et al. (2013). Pandora total column $NO_2$ varied by an order of magnitude spatially and temporally with distinct diurnal and weekly patters in polluted areas. The $NO_2$ column average difference between Pandora and OMI ranged from -0.17 DU to +0.05 DU with OMI mostly showing underestimates, particularly in urbanized areas where pollution
sources were located close to measurement sites. For total column $O_3$ the average differences were <12 DU (or 3.9%) with OMI greater than Pandora. Reed et al. (2015) also compared Pandora with OMI during the Baltimore/Washington campaign and found that OMI pixel size, clouds, and aerosols affected OMI retrievals causing differences between Pandora and OMI of up to 65% for total column $NO_2$ and 23% for total column $O_3$. Tzortziou et al. (2012) examined total column $O_3$ from Pandora from several sites outside the DISCOVER-AQ domains, such as Cabauw, Netherlands; Helsinki, Finland; and
Fairbanks, Alaska. The average difference between OMI and Pandora was <1% when the OMI pixel was within 50 km of



the Pandora location and cloud fraction was <0.2. The best comparison was at Cabauw where the average difference was 0.3 DU.

The Airborne Compact Atmospheric Mapper (ACAM) flew on the NASA UC-12 aircraft during the Baltimore/Washington DISCOVER-AQ campaign, yielding high-resolution (1.5 x 1.1 km$^2$) DOAS NO$_2$ retrievals for

columns below the aircraft. Slant columns are converted to vertical columns (Lamsal et al., 2016) using NO$_2$ profiles from a high-resolution regional air quality model and bidirectional reflectivity distribution function data from MODIS. ACAM retrievals compared well with vertically-integrated in-situ data. Model tropospheric columns above the aircraft were subtracted from the OMI tropospheric column retrievals and the remainders were compared with the ACAM data (see Fig. 17). ACAM demonstrated intra-urban spatial variability not possible with OMI, with factor of four subpixel variability seen

within some OMI pixels.

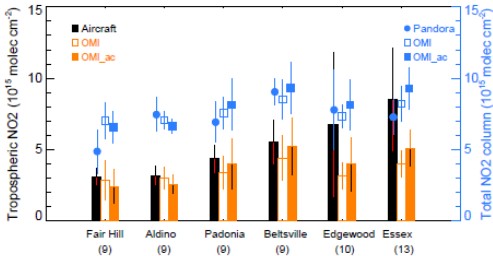

**Figure 17: Comparison of average tropospheric (orange bars) and total (blue squares) NO2 columns determined from in situ aircraft (black bars) measurements and total columns retrieved from Pandora (filled blue circles) at the**
**six locations in Maryland during the DISCOVER-AQ field campaign (from Lamsal et al., 2014). Open bars and squares represent the operational retrievals and filled bars and squares represent the retrievals performed using collocated aircraft-measured NO2 vertical profiles. The vertical lines represent the standard deviation of the average.**

Ozonesondes were launched typically twice a day during DISCOVER-AQ flight days from two sites in the
Baltimore/Washington area. Total column O$_3$ from the sondes averaged 3% greater than OMI columns (Thompson et al., 2014), which considering the 5% sonde uncertainty is not statistically significant. The sonde data were also compared with trajectory-mapped OMI minus MLS tropospheric O$_3$ residual (surface to 200 hPa; Schoeberl et al., 2007), yielding a statistically significant average discrepancy of 10%.

The NO$_2$-sonde developed at KNMI is a unique, lightweight instrument that can be readily deployed on a variety of
platforms (Sluis et al., 2010). During DISCOVER-AQ the NO$_2$-sonde was operated on a tethered balloon platform and used to fill a critical gap between the ground level and the lowest extent the NASA P3-B aircraft that regularly made spiral flights over the NO$_2$-sonde site. The NO$_2$-sonde data from the tethered balloon can be combined with the NO$_2$ measurement from the NCAR NOxyO3 instrument on board the P3-B as both use chemiluminescence and have a sampling rate of 1-Hz. These combined vertical profiles of NO$_2$ in the boundary layer and lower portion of the free troposphere offer the opportunity to
test the assumptions of NO$_2$ profile shape that are used in both models and the OMI retrieval of NO$_2$.

Using aircraft and balloon pressure data, the sonde and aircraft datasets were collocated in altitude. The tethered NO$_2$-sonde measured profiles from the ground surface up to 500 meter above ground level (AGL). The aircraft conducted 2 to 3 descending profiles over the sonde site. These datasets were used in two ways to examine OMI NO$_2$ retrievals. The combined aircraft-sonde data was first used to create a pseudo-column for comparison with OMI and Pandora located at the
Huron, CA site in January 2013. Second, a combined sonde-aircraft profile was created for the afternoon profile taken on 22 Jan 2013 compared to the a priori model profile shape to analyze the possible error in assumed profile shape as compared to





the 'actual' profile shape measured by the sonde and aircraft. The well-mixed afternoon boundary layer in this case led to a small calculated error in column amount of 8% as shown in Fig. 18.

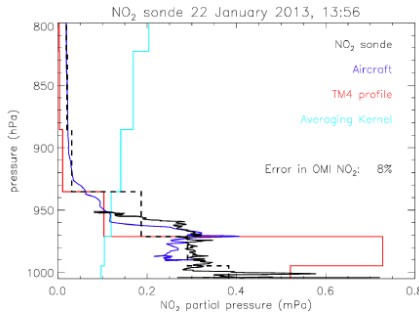

**Figure 18: Comparison of NO₂ vertical profile obtained from KNMI NO₂-sonde to in situ aircraft profile, TM4 model profile and OMI (DOMINO) averaging kernel for OMI pixel covering DISCOVER-AQ site Huron, California on 22 January 2013.**

The combined NO₂-sonde-aircraft profiles provide a detailed look at highly dynamic NO₂ concentrations in the boundary layer and lower portion of the free troposphere. The temperature data from both the tethersonde and the aircraft is also useful for the identification of inversion layers and stable layers which often coincide with enhancements in NO₂ concentration in the lowest, near-surface portion of the boundary layer. Typically, early morning profiles (07:00 – 10:30 LT) exhibited much more vertical variability and heterogeneity, often with multiple layers of enhanced NO₂ concentration (5 to 20 ppbv) as

compared to more well-mixed afternoon profiles (12:00-15:00 LT). With the exception of the earliest morning balloon profiles, the tethersonde did not measure outside of the boundary layer.

The altitude of observed enhanced layers of NO₂ between the aircraft data and the balloon borne NO₂-sonde was consistent within tens of meters. Calibration of the NO₂-sonde to on site reference method NO₂ monitoring instruments ensured close agreement in the absolute concentration of NO₂ on the order of 1-5 ppbv as compared to the aircraft

measurements. Some variation is to be expected given the temporal (up to 25 minutes) and spatial distance (up to 1km) between the aircraft and the NO₂-sonde.

**10.2 CINDI and CINDI-2**

For the validation of space borne observations of NO₂ and other trace gases from hyperspectral imagers, ground based instruments based on the MAXDOAS technique (Honniger et al., 2004; Wittrock et al., 2004) are an excellent choice, since

they rely on similar retrieval techniques as the observations from orbit. In both cases, retrievals take into account the light path of scattered sunlight though the entire atmosphere. Since MAXDOAS instruments are relatively low cost and can be operated autonomously almost anywhere, they are credible candidates to form a world-wide ground based reference network for satellite observations. To ensure proper traceability of the MAXDOAS observations, a thorough intercomparison is mandatory, which is one of the goals of both the Cabauw Intercomparison of Nitrogen Dioxide Measuring Instruments

(CINDI) and CINDI-2 campaigns.

The Cabauw Experimental Site for Atmospheric Research (CESAR) (Apituley et al., 2008) site in center of The Netherlands was the stage of the CINDI in June-July 2009 (Piters et al., 2012). Cabauw was chosen because the flat terrain offered a free view of large parts of the horizon, needed to accommodate the viewing geometry of the MAXDOAS observations. The location is under influence of both clean as well as polluted air masses. This gives a wide range of possible





trace gas concentrations and mixtures. Furthermore, at CESAR a wide range of observations are routinely carried out that fulfill the requirement to provide the background necessary to unravel the differences between the observations from different MAXDOAS instruments that can be quite diverse in design and data treatment. These observations include observations to understand the light paths, i.e. in-situ aerosol observations of optical and microphysical properties, as well as

vertical profiles of aerosol optical properties by (Raman) lidar (Apituley et al., 2009; Donovan and Apituley, 2013; de Haij et al, 2007). In addition, vertical profiles of $NO_2$ were measured during CINDI using the then newly developed $NO_2$-sonde (Sluis et al., 2010), and a $NO_2$ lidar system (Volten et al., 2009). This approach proved to be highly successful and results were described in papers collected in a special issue (Roscoe et al., 2010). Although no direct OMI validation was performed using data collected during CINDI, the campaign was crucial in establishing the performance and requirements of

the ground based instruments (e.g. Lamsal et al., 2014; Irie et al., 2012).

With the imminent launch of Sentinel-5 Precursor/TROPOMI, with a nadir pixel size of 7.0 x 3.5 $km^2$, and recent developments in MAX-DOAS instruments (e.g. Ortega et al., 2015) there was a need for a renewed MAXDOAS intercomparison campaign: CINDI-2, which was completed in September 2016 and had the goals 1) to assess the consistency of slant column measurements of key target species ($NO_2$, $O_3$, $O_4$ and HCHO) relevant for the validation of

TROPOMI and the future ESA atmospheric Sentinels, from a large number of DOAS and MAXDOAS instruments from all over the world, 2) to study the relationship between remote-sensing column and profile measurements of those species and reference measurements of the same species, and 3) to investigate the horizontal representativeness of MAXDOAS measuring systems in view of their use for the validation of satellite tropospheric measurements on the scale of 25-50 $km^2$. During CINDI-2, 36 MAXDOAS instruments participated. A feature of recent MAXDOAS developments is the ability to

use azimuthal scanning, in addition to elevation scanning such as in e.g. the Pandora type of instruments (Herman et al., 2009).

To support the campaign goals, $NO_2$ profiles were again provided by $NO_2$-sondes and lidar, as well as through in-situ observations and other ancillary observations situated in and around the Cabauw meteorological tower as depicted in Fig. 19. Extensive aerosol information was gathered using Raman lidar as well as by in situ samplers. The analysis of the CINDI-2

data is ongoing.

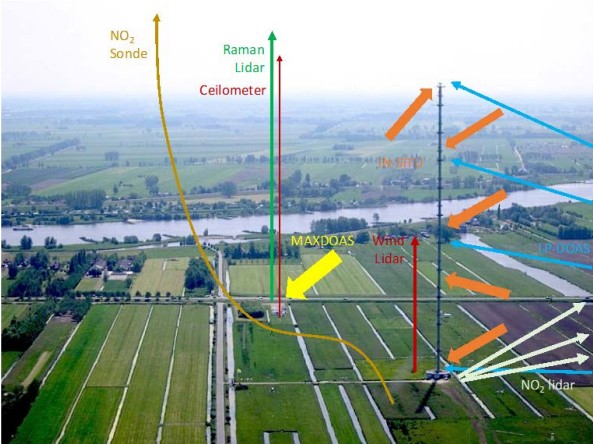

**Figure 19: Schematic layout of the CINDI-2 campaign superimposed on an aerial picture of the CESAR site. Main instrument classes are indicated. The $NO_2$-sondes were launched from close to the tower. A long-path DOAS was**
**placed at a distance of about 4 km from the tower that was able to scan several retro-reflectors at different altitude levels.**





**11 Conclusions**

OMI was successfully launched on NASA's EOS-Aura satellite on July 15, 2004 and since then delivers a huge amount of satellite data for studying the ozone layer, air quality, and climate change. In this paper, we have given an overview of the instrument's exceptional capabilities and we have highlighted the scientific and operational applications obtained with the

OMI data. Detailed results can be found in the papers in this OMI Special Issue and other publications. Due to the broad user community and worldwide use of OMI data in as well the scientific as operational domain, the complete set of results obtained with OMI data extend far beyond the reach of this special issue.

The OMI instrument is the first in its kind that is able to obtain daily global coverage combined with unprecedented spatial resolution, sensing a high-resolution spectrum per ground pixel. This technique, based on the use of a 2 dimensional

detectors and a unique optical design, is now being used in follow-on satellite instrumentation in the European, American, and Asian space programs. Furthermore, OMI is exceptionally stable, more so than any previous UV-VIS satellite instrument.

In the unique trans-Atlantic collaboration between The Netherlands and Finland and the United States, we were able to obtain a very successful international co-operation that enabled us to learn and improve our different techniques of analyses

and interpret the satellite measurements. In this co-operation the instrument and retrieval knowledge built-up over the last 20-40 years in the USA on one hand and Europe on the other, was exchanged and lead to mutual improvement and understanding of the measurements and its interpretation, with important outcomes for research on the chemical composition of the atmosphere. Especially the development of the retrieval codes with different approaches and application to the same instrument data led to an enormous improvement of our understanding of the data and its accuracy, as well as to several

improvements of the retrieval codes.

The scientific exploitation of OMI data led especially in the air quality domain to new insights and findings, mostly related to the high resolution $NO_2$ and $SO_2$ measurements and their use for air quality forecasts, emission estimations, source attribution, and trend monitoring. Due to the development of the NRT and VFD data streams, the operational use of OMI data was much larger than expected. Examples are the use of OMI NRT data in the EU Copernicus CAMS project for ozone

layer and air quality products, and the use of the NRT and VFD data for the VAACs for aviation rerouting in case of volcanic eruptions.

There were also many positive unanticipated results with OMI data, such as the first ever observations of glyoxal from space, the use of the extremely stable OMI solar irradiance product by the solar irradiance community for monitoring the sun in relation to climate change, the development of high resolution emission source monitoring, and the unexpected strong

correlation of OMI tropospheric $NO_2$ column measurements with in-situ 2 meter altitude ground based sensors in regional domains. Last but not least, now with an almost 13 year data record, the large contribution to the OMI data record will continue to grow in future as the record lengthens due to OMI's stability and owing to its connection with both past and future.

**OMI data access**

There are several locations for accessing and downloading OMI data. The majority of level 1b and level 2(g) and level 3 data products can be obtained via NASA archives including the Goddard Earth Sciences Data and Information Services Center (GES DISC) (https://disc.sci.gsfc.nasa.gov/Aura/data-holdings/OMI) and the accompanying MIRADOR data search tool (https://mirador.gsfc.nasa.gov/). OMI data can also be found at TEMIS (www.temis.nl). Additional, detailed level 1b and

instrument quality information can be found on the KNMI OMI website (http://projects.knmi.nl/omi/research/calibration/instrument_status_v3/). Very-fast delivery products produced by FMI can be found on the SAMPO site (http://sampo.fmi.fi/).





**Acknowledgements**

The authors of this paper wish to thank everyone who has contributed to the successful OMI project (too numerous to mention them all), and especially the international OMI Science Team. The KNMI activities for OMI are funded by the Netherlands Space Office. US investigators acknowledge funding from several NASA programs including the Aura Science

Team (NNH13ZDA001N-AURAST) and the NASA Solar Irradiance Science Team (NNH15CN67C). The Finnish co-authors are thankful for the funding from Tekes and the Finnish Academy, including the recent projects INQUIRE (267442) and ILMApilot (303876) and SPARK.

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
