# Peer review of "The Ozone Monitoring Instrument: Overview of thirteen years in space"

_Atmospheric Chemistry and Physics, 2017_

## Referee Comment (RC1) · Anonymous Referee #1 · 13 Sep 2017

The overview by Levelt et al. gives a nice summary of OMI, its products and applications. The paper is well written. I have a few suggestions below.

The manuscript lists the science questions that OMI was designed to (help) answer. It also gives examples of how these questions are addressed. It would be nice to draw some general conclusions in respective sections on 1) to which extent these questions have been answered, 2) what are left to be better addressed by future missions such as TropOMI and geostationary instruments, and 3) what would be the key challenges for future missions and applications in addressing these questions.

A major accomplishment of OMI products is their air quality applications, and there are a large amount of papers on air quality characteristics, emission constraint, trends and variability, drivers, and impacts. The present manuscript briefly discusses some works,

which are mainly led or co-authored by the authors here and are mainly focused on the US or Europe. It would be important to expand the discussion to incorporate studies from other researchers and for other parts of the world (especially China and India, the top two populated and polluted countries that have also been heavily studied by using OMI data).

Also, there are a bunch of research products that should be discussed. For example, EOMINO for Europe and POMINO for China are regional NO2 products that specifically address several retrieval limitations in the standard and NRT NO2 products, including treatment of surface reflectance and anisotropy, explicit treatment of aerosol absorption and scattering, etc. This helps a potential OMI user to be aware of a variety of products (and their limitations) provided by the whole OMI community, and allows the user to choose the most suitable product for his/her applications.

---

## Referee Comment (RC2) · Anonymous Referee #2 · 9 Oct 2017

General Comments:

This is an excellent overview of OMI performance and results, and represents a monumental undertaking to summarize the efforts of many researchers taken over the last 13 years. OMI is an important bridge between the US BUV type instruments and the European GOME/SCIAMACHY instruments to the yet to be flown atmospheric composition instruments. The paper opens with the stated science objectives of the OMI instrument, followed by a review of the instrument features and performance, and then summarizes the many science results and applications by operational users. The paper is an excellent complement to this ACP special issue and provides the reader, with the more than 200 references, an excellent resource for OMI capabilities.

Although the paper has many contributors, the detailed content reads fluently and with

an even technical level. There are other ways to organize large number of topics in the paper, but the organization as presented is appropriate.

Specific comments:

Since all the (that this reviewer can tell) science findings and technical results that appear in this paper have been previously published, there are no comments that challenge or question the results.

Section 2.2 deals with air quality forecasting. On page 10, line 11 what is the connection with "..total ozone column have been assimilated. . ."?

Page 11. What is the significance of Figure 3? What is the quality of the comparison, and what is the conclusion?

Page 12. What is to be concluded from Figure 4? The data are for the tropics, not an important source of pollution compared to Northern mid latitudes. This section deals with pollution trends. Perhaps these results pertain more to climate.

Page 13, first paragraph may better fit in section 7.3 that deals with synergy by combining OMI and other instrument data.

Page 18, SSI Monitoring is very important topic, but most of the discussion has been published and repeated here. The level of detail is out of proportion for this review paper.

Page 23, line 14. Are the predicted ozone trends detectable with this level of long term accuracy? That is the goal of these efforts and a top level OMI objective.

Page 24, Figure 14. The content of the little green circles at end of each bar is hardly discernable. Explaining what they are might be useful. Deleting them will not reduce the value of the figure. In the caption, what is the basis of the 220 DU?

Page 24, Line 20. Should the MSR-1 results appear in the next section on UV. Where these data used for trend detection? How do they compare with the results in Figure

15?

Page 26, Figure 16. The patterns are similar, but are the differences significant for the purpose the climatology was produced?

Page 27, Line 7. What are the "global correlative datasets"?

Page 28, Section 8.2. $H_2O$ is an important constituent. Are there global maps, trends, more research? For example, Wang et al. "Results show that the OMI data track the seasonal and interannual variability of TCWV for a wide range of climate regimes".

Page 31, Section 9.5. If this section is included in the article, then some mention of results should also be included. This would be consistent with other sections.

Page 32, Section 10. Cal/Val is a major activity for every space mission. For this overview paper, there is too much detail, however. OMI Cal/Val should be a paper in itself, and I would expect that a detailed description and results will appear in this special issue. Highlights only should appear in this review paper.

Technical Corrections:

Page 15, Figure 6 caption. What is "16-18"? Define HTAP. Should the phrase in the parenthesis say "(the sum of HTAP and missing sources)"?

Page 25, Line 7, I could not find "Douglass et al., 2011", "Bais et al., 2007," or the WMO report in the References list. Perhaps they were lead authors in the UV chapters therein.

Page 25, Line 24. The acronym "BAMS" should be defined.

---

## Author Comment (AC1) · 19 Jan 2018

Please, refer to the attached files for the comments to the reviewers and an updated manuscript.

Please also note the supplement to this comment:
https://www.atmos-chem-phys-discuss.net/acp-2017-487/acp-2017-487-AC1-supplement.zip

---

## Author Response (AR1)

**General comment:**

The authors would like to thank both Referee #1 and #2 very much for their time and effort to provide helpful comments and suggestions. These inputs have led to an improved article.

**Response to Anonymous Referee #1**

A response is given to each of the specific comments from Referee #1 here as follows.

**Comment:**
The manuscript lists the science questions that OMI was designed to (help) answer. It also gives examples of how these questions are addressed. It would be nice to draw some general conclusions in respective sections on 1) to which extent these questions have been answered, 2) what are left to be better addressed by future missions such as TropOMI and geostationary instruments, and 3) what would be the key challenges for future missions and applications in addressing these questions. A major accomplishment of OMI products is their air quality applications, and there are a large amount of papers on air quality characteristics, emission constraint, trends and variability, drivers, and impacts. The present manuscript briefly discusses some works, which are mainly led or co-authored by the authors here and are mainly focused on the US or Europe. It would be important to expand the discussion to incorporate studies from other researchers and for other parts of the world (especially China and India, the top two populated and polluted countries that have also been heavily studied by using OMI data).

**Response:**
Section 1.1, the introduction to Section 2, Section 3, and part of Section 6.1 have been rewritten to show how the initial OMI science questions have been answered. To answer the referee's second and third points, additional text has been added to most sections to show how TROPOMI and future missions will able to answer these and new questions in more depth.

Lastly, we have added many new references in Section 2 and 3 to highlight recent results from research conducted in China and India.

**Comment:**
Also, there are a bunch of research products that should be discussed. For example, EOMINO for Europe and POMINO for China are regional NO2 products that specifically address several retrieval limitations in the standard and NRT NO2 products, including treatment of surface reflectance and anisotropy, explicit treatment of aerosol absorption and scattering, etc. This helps a potential OMI user to be aware of a variety of products (and their limitations) provided by the whole OMI community, and allows the user to choose the most suitable product for his/her applications."

**Response:**
Additional text has been added to Section 1.4 to define the differences between OMI standard, research, and regional data products. The EOMINO and POMINO examples of regional dataset have been given and the reader is referred to section 8 for more information on OMI research data products.

**Response to Anonymous Referee #2**

A response is given to each of the specific comments from Referee #2 here as follows.

**Comment:**
Section 2.2 deals with air quality forecasting. On page 10, line 11 what is the connection with "…total ozone column have been assimilated…"?

**Response:**
The following text and reference have been added for clarification, "…using similar methods as described in Inness et al., 2013…".

**Comment:**
What is the significance of Figure 3? What is the quality of the comparison, and what is the conclusion?
**Response:**
Additional text has been added to clarify and further explain the content of Figure 3.

**Comment:**
What is to be concluded from Figure 4? The data are for the tropics, not an important source of pollution compared to Northern mid latitudes. This section deals with pollution trends. Perhaps these results pertain more to climate.
**Response:**
The aim of this subsection is to highlight the OMI data record can be used to evaluate trends. As such, the title of this subsection has been changed to "Trends in Pollutants and Aerosol Presence", and an additional introductory sentence has been added.

**Comment:**
Page 13, first paragraph may better fit in section 7.3 that deals with synergy by combining OMI and other instrument data.
**Response:**
In response to this suggestion, this text regarding combined OMI and TES data has been moved to the section with other multi-instrument analyses (section 9.5).

**Comment:**
Page 18, SSI Monitoring is very important topic, but most of the discussion has been published and repeated here. The level of detail is out of proportion for this review paper.
**Response:**
This section has been shortened.

**Comment:**
Are the predicted ozone trends detectable with this level of long term accuracy? That is the goal of these efforts and a top level OMI objective.
**Response:**
Two additional sentences including extra references have been added to the answer this question. Predicted long term trends in mid-latitude total ozone for the post-peak concentrations of Ozone Depleting Substances (ODS) are of the order of 1%/decade for the period 2000-2015 and are expected to continue to do more or less linearly until the end of the 21$^{st}$ century. See for example, Chipperfield et al. 2017 as shown in their figure 5 and Eyring et al. 2010 as shown in their table 4. A 1% increase in annual mean total ozone due to ODSs combined with a 1% annual mean total ozone column error should lead to a statistically significant trend solely due to decreasing ODSs for a record length of approximately 20 years. For the tropics recovery detection in total ozone columns will take longer as the change in total ozone is smaller. For Antarctica, the detection of recovery should be earlier as the change in total ozone is (much) larger. The large interannual variability at in particular mid-latitudes and to a lesser extent at Antarctica overwhelms the small year-to-year changes in total ozone due to the small year-to-year change in ODSs. Every extra percent total column error adds approximately ten years to the expected mid-latitude recovery date (2% → 30 years, 3% → 40 years, 4% → 50 years).

**Comment:**
Figure 14. The content of the little green circles at end of each bar is hardly discernable. Explaining what they are might be useful. Deleting them will not reduce the value of the figure. In the caption, what is the basis of the 220 DU?
**Response:**
Both the figure and caption have been updated for clarity and readability.

**Comment:**
Page 24, Line 20. Should the MSR-1 results appear in the next section on UV. Where these data used for trend detection? How do they compare with the results in Figure 15?
**Response:**
The wording has been slightly modified and this sentence including new reference has been moved to the end of section 6 to tie in with UV data discussion.

Comment:
The patterns are similar, but are the differences significant for the purpose the climatology was produced?
**Response:**
Additional text has been added in the second paragraph of section 7 and the caption for Figure 16 (now Figure 15) has been rewritten to address this question.

**Comment:**
Page 27, Line 7. What are the "global correlative datasets"?
**Response:**
To clarify, the term "global correlative datasets" has been removed and replaced with the following text:
"… agrees globally well with high vertical resolution limb viewing satellite observations (including MLS, TES, SAGE-II, GOMOS, OSIRIS, HALOE) and ozone soundings…".

**Comment:**
Page 28, Section 8.2. $H_2O$ is an important constituent. Are there global maps, trends, more research? For example, Wang et al. "Results show that the OMI data track the seasonal and interannual variability of TCWV for a wide range of climate regimes".
**Response:**
To address this question the text has been expanded with additional text and references including Wang et al.

**Comment:**
Page 31, Section 9.5. If this section is included in the article, then some mention of results should also be included. This would be consistent with other sections.
**Response:**
This section has been expanded with additional results and related references.

**Comment:**
Page 32, Section 10. Cal/Val is a major activity for every space mission. For this overview paper, there is too much detail, however. OMI Cal/Val should be a paper in itself, and I would expect that a detailed description and results will appear in this special issue. Highlights only should appear in this review paper.
**Response:**
The section has been shortened with some parts rewritten to present more of an overview. OMI validation in the early stages of the mission has been thoroughly summarized in Schoeberl et al.,

2008 and is now cited in the text. However to highlight the benefit of recent innovative campaigns with measurement approaches relevant to OMI, one North American campaign, DISCOVER-AQ and one European campaign are described in more detail.

A response is given for each of the suggested technical corrections from Referee #2 here as follows.

**Comment:**
Figure 6 caption. What is "16-18"? Define HTAP. Should the phrase in the parenthesis say "(the sum of HTAP and missing sources)"?
**Response:**
The caption has been updated to define HTAP and clarify the text.

**Comment:**
Page 25, Line 7, I could not find "Douglass et al., 2011", "Bais et al., 2007," or the
WMO report in the References list. Perhaps they were lead authors in the UV chapters therein.
**Response:**
These references have been added to the reference section.

**Comment:**
Page 25, Line 24. The acronym "BAMS" should be defined.
**Response:**
The BAMS acronym was listed as a part of a reference that has been updated and now appears in the text and reference section as follows:

[revised manuscript text omitted]

Stein Zweers, Debora…, 13/1/2018 15.38

Stein Zweers, Debora…, 13/1/2018 15.38

Stein Zweers, Debora…, 13/1/2018 15.38

Stein Zweers, Debora…, 11/1/2018 12.04

Stein Zweers, Debora…, 11/1/2018 12.05

Stein Zweers, Debora…, 11/1/2018 12.05

Stein Zweers, Debora…, 11/1/2018 12.06

Stein Zweers, Debora…, 11/1/2018 12.06

Stein Zweers, Debora…, 11/1/2018 12.07

Stein Zweers, Debora…, 11/1/2018 12.07

Stein Zweers, Debora…, 11/1/2018 12.08

Stein Zweers, Debora…, 11/1/2018 12.08

Stein Zweers, Debora…, 11/1/2018 12.08

Stein Zweers, Debora…, 11/1/2018 12.08

Stein Zweers, Debora…, 13/1/2018 16.12

Stein Zweers, Debora…, 11/1/2018 12.22

Stein Zweers, Debora…, 11/1/2018 12.22

Stein Zweers, Debora…, 11/1/2018 12.23

Stein Zweers, Debora…, 13/1/2018 16.13

Stein Zweers, Debora…, 13/1/2018 16.08

Stein Zweers, Debora…, 11/1/2018 12.13

Stein Zweers, Debora…, 12/1/2018 18.15

[revised manuscript text omitted]

Johanna Tamminen 9/2/2018 13.58

Stein Zweers, Debora…, 11/1/2018 12.27

Stein Zweers, Debora…, 13/1/2018 16.16

Stein Zweers, Debora…, 13/1/2018 16.17

Stein Zweers, Debora…, 11/1/2018 12.28

Stein Zweers, Debora…, 13/1/2018 16.15

Stein Zweers, Debora…, 13/1/2018 16.15

Stein Zweers, Debora…, 11/1/2018 12.28

Stein Zweers, Debora…, 13/1/2018 16.15

Stein Zweers, Debora…, 11/1/2018 12.28

Stein Zweers, Debora…, 13/1/2018 16.15

Stein Zweers, Debora…, 11/1/2018 12.29

Stein Zweers, Debora…, 13/1/2018 15.53

Stein Zweers, Debora…, 11/1/2018 12.29

Stein Zweers, Debora…, 13/1/2018 15.47

Stein Zweers, Debora…, 11/1/2018 12.30

[revised manuscript text omitted]